



# Dome instability at Merapi volcano identified by drone photogrammetry and numerical modeling

Herlan Darmawan[1,2*], Thomas R. Walter[1], Valentin R. Troll[3,4], Agus Budi-Santoso[5]

[1]Dept. Physics of Earth, GFZ German Research Center for Geosciences, Telegrafenberg, 14473, Potsdam, Germany.
[2]Laboratory Laboratory of Geophysics, Department of Physics, Faculty of Mathematics and Natural Sciences, Universitas Gadjah Mada, Yogyakarta, Indonesia.
[3]Dept.of Earth Science, Section for Mineralogy, Petrology and Tectonics, Uppsala University, Villavägen 16, SE-752 36 Uppsala, Sweden.
[4]Faculty of Geological Engineering, Universitas Padjajaran, Jatinangor 45363, Bandung, Indonesia.
[5]BPPTKG (Balai Penyelidikan dan Pengembangan Teknologi Kebencanaan Geologi), Jalan Cendana 15, Yogyakarta 55166, Indonesia.

*Correspondence to*: Herlan Darmawan (herlan@gfz-potsdam.de; herlan_darmawan@ugm.ac.id)

**Abstract.** The growth of lava domes may cause gradual oversteepening and can lead to gravitational instability and eventual collapse to produce pyroclastic flows that may travel up to several kilometers from a volcano's summit. At Merapi volcano, Indonesia, pyroclastic flows are a major hazard, frequently involving high numbers of casualties. After the VEI 4 eruption in 2010, a new lava dome developed on Merapi volcano and was structurally destabilized by six steam-driven explosions between 2012 and 2014. Previous studies revealed that the explosions produced elongated open fissures and a structurally delineated sector at the southern part of the dome complex. Here, we investigate the geometry, thermal fingerprint, and hazard potential of the delineated unstable dome sector by integrating drone-based geomorphologic data and forward-looking thermal infrared images. The sector located on the un-buttressed southern flank of the steep dome that is delineated by a horseshoe-shaped structure and we identify intense thermal and fumarolic activity along this structure, hosting the high temperatures of the current dome. From the morphology, structures, and thermal mapping, we conjecture that the horseshoe shaped structure may develop into a failure plane that could lead to gravitational collapse of the unstable dome sector. To further elaborate on this instability hypothesis, we calculate the factor of safety, and run a numerical model of the resulting block and ash flows depositional area using Titan2D. Results of factor of the safety analysis confirm dome instability, especially during typical rainfall events. The titan2D model suggests that a hypothetical gravitational collapse of the delineated unstable dome sector would travel southward for up to 4 km distance. This study highlights the relevance of structural development of lava domes, which can affect hazards even years after dome emplacement, and influences the development of thermal and fumarolic activity of cooling lava domes.



**Keywords**: Merapi volcano, unstable dome sector, geomorphology, thermal mapping, factor of safety, pyroclastic hazard.

## 1. Introduction

Lava domes are viscous lava protrusions that accumulate at volcanic vents and experience extrusive (exogeneous) and intrusive (endogeneous) growth (Hale, 2008). During formation of lava domes, they may start lateral flow as coulees, be subject to cooling, subsidence, and often develop concentric structures on the flat-topped summit of the dome (Walter et al., 2013b;Salzer et al., 2017;Rhodes et al., 2018). As these studies show, many details of the development, geometric organization and actual forming processes of the dome structures remain poorly understood. Other factors such as intense rainfall, hydrothermal alteration, gas overpressure, mechanical weakening, and earthquake activity may also augment instability and promote failure of a lava dome (Voight and Elsworth, 2000;Reid et al., 2001;Ball et al., 2015). Once fracture arrangements are established in a lava dome, rain water is able to percolate, causing mechanical weakening associated with hydrothermal alteration and gas overpressure that is potentially leading to dome collapse even during periods of quiescence (Voight and Elsworth, 2000;Reid et al., 2001;Elsworth et al., 2004;Simmons et al., 2004;Taron et al., 2007;Ball et al., 2015). Lava domes that become structurally destabilized by these processes may cause hazardous block and ash flows (Calder et al., 2015). The dome collapse at Soufrière Hills Volcano (SHV), Montserrat, is an example of rain-triggered collapse that followed a period of quiescence in 1998-99. The SHV dome collapse produced pyroclastic density currents (PDC) with a volume of $22\times10^{6}$ m$^{3}$ that traveled for a distance of ~3 km (Norton et al., 2002;Elsworth et al., 2004). The rain water infiltrated through identified fractures and produced gas overpressure within the lava dome carapace, which resulted in collapse of those regions that were characterized by hydrothermal alteration and weakened structures (Voight, 2000;Elsworth et al., 2004). This event demonstrated that improvement of our ability to assess the instability of lava domes is crucial for volcanic hazard mitigation.

Identifying the potential hazard of lava domes is often difficult and requires high quality observational data, complemented by modelling analyses (Voight, 2000). Dome building volcanoes are often steep sided hazardous edifices, where direct field access is very limited and acquisition of high quality field data challenging. Remote sensing techniques, such as satellite imageries, aerial photogrammetry and thermal imaging can provide detailed information on the structure, deformation, morphology, and thermal signature of active lava domes (James and Varley, 2012;Walter et al., 2013a;Salzer et al., 2014;Thiele et al., 2017;Darmawan et al., 2018), which allows to study important parameters for assessment of dome instability and potential hazard (Voight and Elsworth, 2000;Elsworth et al., 2004;Simmons et al., 2004;Taron et al., 2007). As geomorphology and thermal mapping may indirectly allow identifying structural lineaments, fractures and slope changes, their combination is relevance for investigating the structural stability of lava domes. Based on previous studies, the degree of instability of lava domes can be, to first order, assessed by the factor of safety (Voight and Elsworth, 2000;Simmons et al., 2004;Taron et al., 2007). Lava dome collapses, in turn, can be simulated by mass flow software such as Titan2D (Patra et al., 2005;Sheridan et al., 2005), which has been used to map dynamics and distribution of block and ash flows at Merapi before (Charbonnier and Gertisser, 2009, 2012;Charbonnier et al., 2013).



In this study, we present a comprehensive assessment of dome instability and the potential hazard of the current Merapi lava dome complex. We first provide a short summary about the eruption history and dome formation. Subsequently, we introduce the topographic and thermal data used and the methods utilized to identify fractures and the dimension of the unstable dome sector. This information allows us to analyze the factor of safety, and to set up forward simulations of the Titan2D software. Results allow an assessment of Merapi dome instability, and thermal and structural weakening of domes in general. The combined results help to understand the relevance of dome fracturing and to outline the potential hazard zone affected in case of a dome sector collapses.

### 1.1. Merapi volcano

Merapi volcano is a basaltic to andesitic volcano that formed due to subduction of the Indo-Australian oceanic plate beneath the Eurasian continental plate (Hamilton, 1979). Merapi volcano is one of the most active and dangerous volcanoes in Indonesia, with more than 1 million people living on the volcano's flanks, and the city of Yogyakarta with 3 million inhabitants is located only ~30 km distant from the volcano's summit (Fig. 1) (Lavigne et al., 2015). The volcanic activity of Merapi is well documented since the 1800's and its typical eruption style is dome extrusion and block and ash flows (Voight et al., 2000a). The extrusion rate of a lava dome at Merapi may strongly vary, ranging from ~0.04 $m^3$/s (Siswowidjoyo et al., 1995), up to 35 $m^3$/s or more during volcanic crisis (Pallister et al., 2013).

Merapi shows signs of interactions with surrounding environmental influences, and for instance rainfall appears to correlate with fumarole activity and seismic intensity (Richter et al., 2004), and tectonic earthquakes influence eruptive activity (Walter et al., 2007;Walter et al., 2015;Carr et al., 2018). The volcano erupted several times during the last decades, on average once every 3-5 years, with the largest explosive event recorded in 2010. The 2010 eruption removed parts of the summit area (Surono et al., 2012), excavated a ~200 m deep crater and was followed by regrowth of a new dome (Kubanek et al., 2015). The new lava dome was intermittently destroyed by several explosive events again between 2012 and 2014, which also caused elongated open fissures (Fig. 1b and c) (Walter et al., 2015), and produced a structurally delineated dome sector located at the southern part of the dome (Fig. 1c) (Darmawan et al., 2018). This dome sector is posing a hazard and may collapse, possibly associated with rainfall events and which is the focus of this study.

## 2. Data and method

### 2.1. Observational data

Geomorphological and thermal infrared data were acquired in the years after the 2012-14 explosive events. The geomorphological analysis is based on a high resolution Digital Elevation Model (DEM) and a photomosaic. The high resolution digital elevation model was generated by merging the 3D point clouds of a Terrestrial Laser Scanning (TLS) and of a Structure-from-Motion (SfM) photo-drone dataset acquired in September 2014 and October 2015, respectively.





The TLS data was acquired by Riegl 6000 instrument from the eastern rim of the summit crater (latitude = 7° 32′ 25.0161″ S, longitude = 110° 26′ 51.2110″ E), looking down westward onto the dome. We could gather about 2.8 Mio data points of the inner crater area including the dome. For the TLS setup, we were using a Pulse Repetition Rate (PRR) of 30 kHz, an observation range of 0.129–4393.75 m, a theta range (vertical) of 73–120° and a sampling angle of 0.041°, and a phi range (horizontal) of 33°–233° with a sampling angle of 0.05°. We used 12 local reflectors to correct for rotation errors. A major benefit of the TLS methodology is the high resolution and precision in the field of view, however shadowing effects were significant. Therefore we added a SfM dataset, derived from drone photographs which we acquired by camera quadcopters (DJI Phantom 3 and DJI MavicPro). The DJI Phantom drone flew loops at a height of ~ 140 m over the dome and took nadir photographs at regular interval of 2 second with a 12 megapixel resolution. These photographs were processed by the Structure from Motion and Multi View Stereo (SfM-MVS) workflows (Szeliski, 2011). We then combined the TLS and SfM datasets by point pair-picking registration in the Cloud Compare software. More details about the data acquisition, TLS and SfM data processing are described in full detail in Darmawan et al. (2018). From the combined 2014-2015 data of around 58 million data points located in 3D space, georeferenced in WGS 84 frame, we generated a gridded Digital Elevation Model with a resampled constant resolution of 0.5 m. This digital elevation model was used for geomorphological analysis. In addition, the drone photos allowed generation of a photomosaic, which helped identifying structures, fractures and fumarole sites. While the TLS dataset and data handling described here is similar to the one in Darmawan et al. (2018), we also added a new photo-drone dataset.

The new photo-drone dataset was acquired using DJI Mavic pro camera drone with resolution of 12 Megapixels in 2 September 2017, flew only ~50 meters above the dome. Aim of this new dataset was to test if any changes have occurred at the studied unstable dome flank. We captured around 408 aerial images that were used for photogrammetry analysis. However, as strong degassing at the fumaroles limited visibility, 3D reconstruction by using the SfM-MVS technique was found to be less effective. Therefore, the aerial images acquired in 2017 were only qualitatively compared to the previously described 2015 aerial images (published in Darmawan et al. (2018)) to explore possible changes in structure and fumarole activity. The camera drone data allows identifying fractures and the geometry of the delineated dome sector that are used as parameter in factor of safety calculation. We use ArcMap to grid the data and generate hill shade and slope maps, allowing identifying fracture arrangements and quantifying the dimension of a structurally delineated dome sector. This data provides geometric information for estimating the stability of the dome sector.

As we are mapping the structural architecture of the southern dome sector, we are also interested in their role in fluid transport and fumarole activity. Fractures and lithology contrasts may lead to permeability differences that control the pathways for thermal fluids (Ball and Pinkerton, 2006). We recorded the apparent temperature distribution of the Merapi lava dome by using a forward looking infrared (FLIR) P660 thermal camera in September 2014. Images were taken from the eastern crater rim close to the TLS station (Fig. 1b). The FLIR camera operates on a spectral band of 7.5 – 13 µm which allows us to identify an apparent temperature range which was calibrated in a range 0 – 500°C. The resolution of the FLIR cameras is 640 × 480 pixels. The FLIR camera was equipped with a 7° ($f = 131$) zoom lens with a 0.38 mrad instantaneous





field of view (Walter et al., 2013a), allowing generation of very detailed and high resolution thermal images, with estimated pixel dimensions of 1 px= 0.05 m on the dome center. We combined over 25 such images and generated a thermal mosaic, allowing retrieval of high resolution apparent temperature maps.

Thermal infrared data is dependent on a number of environmental parameters, such as the distance and emissivity of the
target (the dome), the solar reflection, the viewing angle, the atmospheric effect, and the presence of particles/gases in the electromagnetic radiation path (Spampinato et al., 2011). We recorded the thermal images during night time (5 am local time), so that background temperature was low, and insulation artifacts and solar reflection were minimized. Other factors were solved in data processing by setting the emissivity and transmissivity values to 0.98 and 0.7, respectively, following Carr et al. (2016) and Ball and Pinkerton (2006). Relative humidity was set to 45% according to weather observation. The
relative distance to the dome was on average 300 m and the background temperature was assumed to be 10 °C. After defining the parameters, the thermal images were set to constant color scales for all images, and were then mosaicked to obtain a high resolution panorama image of the apparent thermal distribution of the Merapi lava dome.

## 2.2. Factor of safety (FS)

Slope instability can be assessed deterministically by estimating the load carrying capacity of a flank of the Merapi dome. The factor of safety describes if a system is stronger or weaker for the given load, which is also affected by rainfall, and has been applied for numerous engineering problems (Aleotti and Chowdhury, 1999). At dome building volcanoes, the factor of safety calculation allows estimating slope instability during precipitation events, as dome collapse events may be associated with heavy rainfall (Yamasato et al., 1998;Elsworth et al., 2004). Here we follow the work of Simmons et al. (2004) and test
the instability of the southern sector of the Merapi lava dome during intense rainfall. We first estimate how deep rain water is able to percolate ($d$) through identified fractures:

$$d = \frac{is^2}{4K_R} \frac{\rho_w c_w}{\rho_R c_R} \frac{\Delta T_w}{\Delta T_R} \times 1.13\sqrt{t_D} \qquad (1)$$

Where the dome sector has a thickness $h$, and is inclined at $\alpha$, and mechanically isolated by a fracture of spacing, $s$ (Simmons et al., 2004;Taron et al., 2007). In our case the fracture spacing is determined the dimensions of the unstable dome sector and is measured by our geomorphological data. Rainfall $i$ is the intensity/magnitude as measured at a proximal weather station,
and $\Delta T_w$ is the required thermal energy to vaporize water, $\Delta T_R$ is the required thermal energy to cool the fracture surface, $\rho_R$ is the density of lava dome rock, $K_R$ is thermal diffusivity, $t_D$ is a non-dimensional time which described as $= K_R.t / l^2$, $t$ is the rainfall duration, $l = $ s/2, $c_W$ and $c_R$ are heat capacity of water and rock, respectively.

The estimated water percolation ($d$) is then used to calculate the factor of safety:

$$Fs = \frac{C + (W \times cos(\alpha) - (Fu)) \times tan(\emptyset)}{W \times sin(\alpha) + Fw + Fv} \qquad (2)$$

where $C$ is the cohesive strength, $W$ (unstable dome sector weight) $= s \times h \times \rho_R \times g$, and $s$ is the fracture spacing, $h$ is the
unstable dome sector thickness, $\rho_R$ is the density of lava dome rock, and $g$ is the gravitational force. During intense rainfall,





the instability is influenced by uplift force from the volcanic gas ($Fu = 0.5{\times}d{\times}cos(\alpha){\times}\rho_w{\times}g{\times}s$ ), water force ($Fw = 0.5{\times}d^2{\times}cos(\alpha){\times}\rho_w{\times}g$), and vaporized water force ($Fv = d{\times}cos(\alpha){\times}\rho_w{\times}g{\times}(h\text{-}d)$) (Fig. 6a), where $d$ is the rain water percolation, $\alpha$ is the inclination of failure plane, $\emptyset$ is friction angle, and $\rho_w$ is the density of water. A result of FS $\leq$ 1 indicates a potential failure, whilst for a FS larger than 1 the dome sector is assumed to be stable.

5    Factor of safety calculation requires careful parameter justification. For the parameters, we considered the rain gauge data as recorded at hydro-meteorological stations around Merapi volcano, and set the rainfall intensity ($i$) of our FS model to 10-100 mm/h. The fracture spacing ($s$) is defined to be 100 m that mimics a rotational listric fault with thickness (h) of ~40 m (Fig. 6a). The temperature to cool the fracture from the surface to the dome interior ($\Delta T_w$) was set at 200 - 800°C, which is based on our thermal data and thermodynamic models of lava dome interior (Matthews and Barclay, 2004). Friction angles from 10   30° to 60° were used, which is on the range of friction for rock on rock material (Barton and Choubey, 1977;Stinton et al., 2004). Density of Merapi rock was set at 2242 kg/m³ (Tiede et al., 2005). The cohesion strength ($C$) was set at 10 MPa, which is on the range of the strength of altered rock (Mayer et al., 2016;Pola et al., 2014). Details on the parameters used to calculate the factor of safety and water percolation are listed in table 1 and a critical discussion of the parameters can be found in the section 4.1.

## 2.3. Scenario modeling of block and ash flows

Based on our geomorphologic field analysis and the FS calculations, we are able to simulate a scenario of gravity driven block and ash flows induced by collapse of the unstable Merapi dome sector. We use the Titan2D software and consider collapse of the structurally delineated dome sector only, volumes of which are estimated from our geomorphologic dataset 20   and the failure plane inclination based on the calculation of factor of safety. The Titan2D software has been used by previous studies to simulate block and ash flows due to lava dome collapses (Widiwijayanti et al., 2007;Charbonnier and Gertisser, 2009;Procter et al., 2009;Charbonnier and Gertisser, 2012;Charbonnier et al., 2013). Parameterization is important to generate a model that considers the topography effect during flows simulation. In Titan2D, bed friction is the most sensitive parameter that controls the resulting pyroclastic flow and material distribution (Sheridan et al., 2005;Charbonnier and 25   Gertisser, 2009). Therefore, we used bed friction angles as investigated for the post-eruption Merapi scenario in simulations described in detail in Charbonnier et al. (2013), which have a confidence level of 70%. The bed friction angle was defined between 22° and 8° from the top of the dome at 2805 m elevation to the lowest slope at 1181 m elevation, respectively (Table. 2).

For the scenario simulation, we set the initial velocity of 0 m/s as we assume that the failure mechanism is not involving 30   large magmatic pressure. We further set a maximum volume of $0.3{\times}10^6$ m³, which represents a deep water percolation and gentle slope failure plane (α). To realize the simulations, we used our digital elevation model at the summit region (see chapter 2.1) and extended in the far field by merging the published digital elevation model derived from TanDEM-X data (Kubanek et al., 2015). Our summit region DEM provides a detailed topography model of the current dome, while the



TanDEM-X describes the topography of the flanks of Merapi after the 2010 eruption. A full set of parameters used for Titan2D simulation are listed in table 2.

## 3. Results

### 3.1. Geomorphology and structure of the southern part of the Merapi dome

The high resolution hill shade and slope maps (Fig. 2) provide a detailed morphological and structural view of the Merapi dome. We identify a flat-topped dome that is dissected by a major northwest-southeast trending open fissure and surrounded by large number of bombs and blocks on the top dome, associated with the 2012-14 phreatic explosions (Darmawan et al., 2018). On the southern sector of the dome, the hill shade map further shows that the dome is rather steep, and hosts abundant fractures and a blocky appearance (Fig. 2b). The slope is inclined at ~50° (Fig. 2b), and appears dissected into two or three steep regions, which are separated by gently inclined terraces, which are also seen in profile (Fig 2c). The western region of the southern dome sector is enveloped by a structure that bears a visible morphological expression, being connected to the main northwest-southeast trending open fissure. The upper region of the sector is characterized by a morphologic depression and fracture that resembles a horseshoe-shaped fault-like structure (red line in Fig. 2a), which is open to the south, and can be traced for a length of over 165 meters. Cross section profiles of line p-q and r-s show that the maximum depth of the horseshoe-shaped fault-like structure in the northwest, northeast, and southwest is 6 m, 8 m, and 3 m, respectively (Fig. 2c and d). The horseshoe-shaped structure delineates a dome sector with extension of 100 m x 80 m (Fig 2c and d). As the unstable dome sector is located on a steep slope (Fig. 2b), it is critical to monitor changes at the southern part of the Merapi dome.

Drone cameras recorded close range aerial images and show details of the structures associated with the investigated dome sector (Fig. 3). The structurally delineated dome sector is composed of two blocks, referred to as block I and block II. The surface of the entire dome is highly fractured. Absence of strike slip kinematic indicators suggests that the fractures on the unstable dome sector surface do not show major displacement, possibly involving a significant tensile stress field. Our high resolution DEM further shows that the small fractures at the surface of the unstable dome sector are not very deep, commonly less than ~0.5 m. Based on the dimension of deep reaching fractures that define the block, the main fracture spacing ($s$) is on the order of 100 m (Fig. 2c).

Comparison of the drone images taken in 2015 to those taken in 2017 reveals that minor changes occurred along the horseshoe-shaped structure. We focus on three fractured areas (c-c', d-d', and e-e' in Fig. 3). In 2015, those three fracture areas were not significantly degassing, while two years later, degassing activity has increased (Fig. 3a and b). A closer view to those fractures (1st, 2nd, 4th, and 5th) reveals that they have a width of 0.3 - 1.3 meters and are actively degassing in the 2017 images, while in the 2015 images appear less active. The yellow color surrounding the active fractures indicates sulfur deposition around the fumaroles, being stronger expressed in the 2017 images taken from the fracture number 5 in area C. The unaltered dome material has turned into yellowish sulfuric color within just two years. The drone images further provide details on the kink in slope on the southern side and the morphological depression located on the western side at mid-height





of the dome flank (Fig. 3a and b). This depression has a near circular geometry, is about 15 x 20 m in diameter and is located at the low inclined terrace described in the morphologic data above. The circular depression has likely developed in association with small steam explosions and was thermally expressed in July 2012 (Darmawan et al., 2018). As the depression located at the kink in slope, we conjecture its location being linked to a lithology contrast of the dome (see

Supplementary material 1). The collocation of a kink in slope and sub circular depression features may also provide structural limits relevant for assessing the stability of the southern dome sector.

### 3.2. Thermal variation of the Merapi dome

Forward looking infrared thermal mapping allows identification of the main regions of hydrothermal fluids flow and

therewith to identify structural features that are fluids pathways. We find that the mean apparent temperature at the dome surface is about 6 - 14°C (Fig. 4a). The low temperature of the dome surface is related to our measurements performed at night and the insulating ash deposits that covered the dome during six distinct phreatic explosions that occurred between 2012 and 2014 (Darmawan et al., 2018). Highest temperatures are found at the northern margins of the dome with a maximum temperature of 201.7°C. The high resolution of 1 px= 0.05 m in the 2014 thermal data allowed further

investigation of the horseshoe-shaped structure in more detail. We show the thermal fingerprint of the fractures in three areas, c, d, and e, with a maximum apparent temperature of 161.2°C, 150.2°C, and 30.6°C, respectively (Fig. 4b). Cross section temperature profile of the horseshoe-shaped structure (Fig. 4b) shows a high thermal area in the structure, which indicating a prominent pathway for hydrothermal fluids.

We repeated the thermal mapping campaign of the lava dome three years later (September 2017). The temperature in area c

increased from 30.6 up to ~70°C, which may indicate formation of high thermal fracture and dome instability. However, as the thermal cameras used in 2014 and 2017 are different, the results cannot be directly compared. More details on this repeat thermal mapping can be found in the supplementary material 2.

### 3.3. Factor of Safety results

Assessment of dome instability during intense rainfall first requires to quantify the effect of rain water. Based on a typical rainfall event that accumulates a water column of 10 – 35 mm/h and assuming lasts ~ 3 hours, we calculated the rain water percolation between 10 and 60 meters by using eq. 1 (Fig. 5). The results of the factor of safety (FS) calculations are shown in Fig. 6, where FS ≤ 1 (dot line in Fig. 6b) indicates a potential failure. We find that the factor of safety may remain stable in the 10 m water percolation scenario (black lines in Fig. 6b). The failure plane inclinations at failure mode (FS ≤ 1) are

steeper than the friction angle (α > Ø), which indicate stability. On the other hand, the factor of safety may decrease in the case of a 60 m deep water percolation scenario (red lines in Fig. 6). In this scenario, the failure plane inclinations at failure mode (FS ≤ 1) are 24° and 51° for friction angles (Ø) of 30° and 60°, respectively. The deep water percolation may increase the total driving forces (Fw, Fv, and Fu see Fig. 6a and eq. 2), reduces the friction force, and therefore destabilizes the lava dome sector. The lava dome sector is therefore in a critical failure condition. Calculation of the factor of safety reveals that





the delineated dome sector is particularly unstable during deep water percolation (d ~ 60 m). Therefore we interpret the results that the dome is destabilized due to intense rainfall episodes and subject to potential failure even with basal inclination of only 24°, smaller than the friction angle ($\alpha < \emptyset$). Using this basal inclination, and the aforementioned structural outline of the unstable sector (block I and II), we can quantify the rock volume being unstable during intense precipitation events at $0.3 \times 10^6$ m$^3$.

### 3.4. Scenario numerical model of block and ash flows

The estimated maximum rock volume of $0.3 \times 10^6$ m$^3$ that was reached for the deep water percolation scenario is now used as an input for block and ash flows simulation. Titan2D simulation results show that the material mobilizes down into the valleys at the south-eastern flank of the volcano and will then be deflected by the Kendil hills located 2 km from the summit (yellow triangle in Fig. 7). After southward deflection, the main flow then travels further down the Gendol valley, while some material is deflected to the Woro river valley (Fig. 8). In the first minute, the flow reaches already a distance of 2.45 km from the summit and reaches a maximum velocity of 58 m/s (Fig. 7a). Within 5 minutes, the main flow reaches a distance of 3.3 km from the summit at a maximum velocity of 47 m/s while it continues to travel along Gendol river valley (Fig. 7b). After 10 minutes, the flow has moved 3.5 km from the summit and the front of the flow is now deflected by a cliff at the western part of Gendol river valley (Fig. 7c). The flow finally stops with a maximum run out distance of 4.1 km (Fig. 7d). Most of the materials are deposited at the upstream of Gendol river with a maximum thickness of 13 meters. Our simulation indicates that the zone inundated by the flow due to lava dome collapse is ~1.6 km$^2$ and exceeds ~4 km from the summit. We also simulate the small dome sector collapse scenario with volume of ~$0.15 \times 10^6$ m$^3$ (the block I in Fig. 3a). The result of the pyroclastic block and ash flows due to small dome sector collapse shows maximum distance of ~3.5 km, hazard zone of ~1.3 km$^2$, and maximum deposit thickness of 13 m. A detail map showing the results of a smaller initial block collapse volume is shown in the supplementary material 3.

### 4. **Discussion**

### 4.1. Limitations

We rely on remote sensing techniques to observe the geomorphology, structures, and thermal variation of the Merapi lava dome as access to reach the current Merapi dome is too hazardous. The TLS dataset could be only realized from the eastern crater wall as limited access at the Merapi summit, and therefore the TLS data have significant shadowing effects. Internally this TLS data is highly coherent and unprecedented. In order to fill data holes, we acquired drone photographs and apply the SfM methodology to extract the topographic information. Moreover, the high resolution aerial images acquired by drone are not only used to derive the digital elevation model, but also to visually identify the morphology, degassing sites, alteration zones and therewith the distribution of fractures. However, we note that in 2015 and 2017 the drone was caught by turbulence due to fumaroles activity during data acquisition which may reduce the images quality.



The thermal variation was investigated by using a FLIR camera which has an accuracy of ± 2 °C (2%). Parameters such as emissivity, surface roughness, viewing angle, atmospheric effects, volcanic gas, instrumental errors, solar radiation, and solar heating may affect the pixel value of the FLIR thermal images (Spampinato et al., 2011). The effect of solar radiation and solar heating was largely reduced by acquiring the FLIR data before sunrise. Influence of the other factors was

controlled by parameters of emissivity, transmissivity, relative humidity, distance, and temperature background in the data processing. We tested the sensitivity of these parameters and we found that emissivity is the most sensitive parameter. Increasing emissivity by 0.01 will reduce the apparent temperature by ~1°C. By assuming the range of emissivity between 0.95 and 0.98, common at dome building volcanoes (Merapi and Colima, Mexico) (Walter et al., 2013a;Carr et al., 2016), we infer that our apparent temperature has an uncertainty of ~3°C. For the structural analysis performed, this is an acceptable

effect.

The degree of dome instability is estimated by using the factor of safety calculation, assuming an intense rainfall event similar to the study from Simmons et al. (2004), where the parameters of dome sector geometry (thickness and fracture spacing), temperature, the friction angle, the rock strength, the intensity and duration of the rainfall may influence the result. Our factor of safety analysis is constrained for the southern Merapi dome sector. For this we hypothesize a fracture spacing

(s) of 100 m, thickness (h) of 40 m, cohesive strength of 10 MPa following the studies of rock strength of altered rock from Mayer et al. (2016) and Pola et al. (2014), dome temperature of 200 – 800 °C during typical rainfall at Merapi (intensity of 10-35 mm/h and duration of ~ 3 hours), and friction angles of 30° and 60°.

Our geomorphological analysis, thermal images, and rainfall gauges provide realistic information of the fracture spacing ($s$), temperature to cool the dome ($\Delta T_R$), and rainfall intensity, however, the parameters of dome thickness, rainfall duration, and

temperature to vaporize rainwater ($\Delta T_w$) have some uncertainty. Here, we tested those parameters and found that the rainfall duration is the most sensitive parameter as it influences the depth of water percolation. Doubling the rainfall duration from 3 to 6 hours with intensity of 35 mm/h will increase the water percolation by up to 10 meters, decrease the factor of safety by ~0.09 and reduce the failure plane inclination (α) by 1°, while the dome thickness and temperature to vaporize the rain water ($\Delta T_w$) are not influenced significantly. Doubling the block thickness only reduce the factor of safety by ~0.01 and reducing

the temperature to vaporize the water ($\Delta T_w$) from 100 to 90°C only increase the depth water percolation by 2 meters. By assuming rainfall duration of 12 hours during the rainy season, we estimate the failure plane inclination has an uncertainty of ± 3° that may influence the volume of the collapsing block by ± 65.000 m³.

In a case of dome sector failure, the hazard zone is estimated using the Titan2D software to simulate the resulting pyroclastic block and ash flows. The topography model and dimension of the collapse source represent realistic as it is based on our high

resolution DEM and TanDEM-X. However, the thickness of the dome sector collapse remains speculative and may influence the volume collapse as mentioned before. We also assumed that the collapse scenario does not involve large magmatic overpressure (initial velocity = 0 m/s) which may further influence the flow of the pyroclastic block and ash. The area of potential block and ash flows hazard might change if 1) an eruption increases overpressure as it will give initial velocity ≥



100 m/s, 2) the initial volume of the collapse source is larger (>VEI 1), and 3) if surges are considered that may jump over the Kendil hill, such as seen during the 2010 eruption (Surono et al., 2012;Komorowski et al., 2013).

4.2. Structures at the unstable dome sector

We find that the delineated unstable dome sector at the southern flank of the Merapi lava dome has horseshoe shaped headwall geometry, identified in geomorphology, visible in drone photos, and provides fumarole gas pathways as seen in infrared images. This horseshoe-shaped structure was weakly expressed in data from 2012, but gradually deepened during 6 phreatic explosions that occurred 2012-2014 (Darmawan et al., 2018). The morphology of the unstable southern dome is showing terraces, which correlate with steps in the slope and with the location of deep depressions, one with 15-20 m diameter. The nature of the terraces structures may be associated with the presence of a lithology interface at this level (see drone image in the supplementary material 1). As the terraces are near parallel to the dome top, we speculate that different magma flow units or intrusion or effusion units from distinct dome building stages or events are responsible for this heterogeneity. The 15-20 m diameter depression formed during the explosions in the period 2012-2014. Such an explosion located at the layering interface may hint towards a lithology control of the volcanic gas pathways, similar as at other volcanoes observed (Schöpa et al., 2017), where fumaroles are located at those layers of higher permeability. The permeability of volcanic material may strongly vary within an edifice, and change with time, hence strongly affecting the stability of a volcano (Heap et al., 2015).

The unstable dome sector is delineated by a curved, horseshoe-shaped structure. As we show in this work, this horseshoe-shaped structure is now morphologically expressed by an up to ~6 m deep depression, and provides a pathway for active degassing. Our thermal and UAV aerial images data confirm that the pathway of active degassing is associated with the fracture location, as peak thermal anomalies are found at these structures. Therefore the horseshoe-shaped fracture acts as a pathway for fluids, which has important implications for the instability of the dome sector. Also at other domes, the fractured carapace is associated with thermal expression that is collocated with major fractures (Hutchison et al., 2013;Salzer et al., 2017). Whether these fractures are deep reaching or not, is difficult to quantify accurately. We here assumed the horseshoe shaped fracture to penetrate deep into the dome and transforming to a listric detachment plane. Also, other studies highlight the relevance of fractures at the flat dome tops, which affect the complex cooling and contracting of domes during periods of quiescence (Salzer et al., 2017). Dominant fractures dissecting a dome carapace may lead to strain localizations, which mimic the deformation anisotropy as identified at active faults (Beauducel et al., 2000). Therefore, the identification of the fracture pattern is relevant to understand mechanical anisotropies that affect the cooling and future development of the dome.

4.3. Implications for future failure

We assess the factor of safety and find instability of the southern dome sector if rainwater percolates ≥ 60 m. We hypothesize that the unstable dome sector might collapse in the foreseeable future. Whether this collapse happens during renewed volcano activity, intense rainfall, or by destabilization due to progressive hydrothermal alteration during non-active





repose remains speculative. Observations from other volcanoes show that these factors may trigger block collapse and generates pyroclastic flows (Reid et al., 2001;Norton et al., 2002;Elsworth et al., 2004). Rainfall could trigger water saturation and increases hydrothermal water circulation and so with time also rock alteration. Hydrothermal alteration weakens the structure and might trigger collapse even without renewed magmatic activity (Reid, 2004). An example of such

a catastrophic collapse is the 1998 debris avalanche at Casita volcano, Nicaragua (van Wyk de Vries et al., 2000). Analysis of hydrothermal altered minerals at Casita Volcano suggested that smectite clay in the hydrothermally altered bedrock reduced strength, promoted high water pressurization and destabilized the slope (Opfergelt et al., 2006).

Identifying the location of hydrothermally altered bedrock within a lava dome or volcano edifice requires geophysical, numerical or mineralogical investigations. Finite element numerical models combined with rock alteration index suggest that

an alteration-induced collapse could be located at shallow level or between the talus and the lava dome core (Ball et al., 2015). Our study indicates that the maximum water percolation may reach up to 60 meters during a typical rainfall event at Merapi. This depth agrees with the results of electrical resistivity tomography (ERT), self-potential, and $CO_2$ flux mapping which found a dual hydrothermal system in Merapi (Byrdina et al., 2017). The first hydrothermal system is close to the summit at the depth of 200 m and upwards, which is underlain by a second hydrothermal system located at depth of 1800 m

beneath the summit.

Uniaxial compressive strength (UCS) experiment of lava dome samples reveal a dilatation of dome rock porosity due to shear stress in the conduit (Lavallée et al., 2013;Heap et al., 2015), while thermal expansion and hydrothermal fluid pore pressure may increase the pore pressure, produce micro cracks, and reduce the stability of dome rock (Day, 1996;Farquharson et al., 2016;Gaunt et al., 2016). Therefore, the presence of fumaroles located along fractures may indicate

the mechanical softening of the material, affecting dome stability even after emplacement. This makes a prognosis of potential future collapse challenging.

4.4. Block and ash flow hazard along the Gendol valley

From our simulation of block failure and mobility along the southerly directed valleys, important implications for hazard

assessment can be drawn. The typical mechanism of pyroclastic block and ash flows generation at Merapi is gravitational dome collapse due to new magma intrusion (Newhall et al., 2000;Voight et al., 2000a), but also external trigger was repeatedly observed at Merapi such as intense rainfall (Ratdomopurbo and Poupinet, 2000), gas overpressure (Voight et al., 2000b), and earthquake (Carr et al., 2018;Walter et al., 2007;Richter et al., 2004). During ascent, magma is fragmented, releases gas, and the physical behavior changes from ductile to brittle (Cashman and Bettina, 2015). When recharging

magma reaches a shallow-level beneath the dome, rapid decompression occurs, which liberates gas and can produce gas overpressure to trigger dome collapse during an eruption (Calder et al., 2015). As the dome material is brittle, the eruption products of resulting pyroclastic flows consequently contain a mixture of materials that range from discrete blocks to fine ash (Branney and Kokelaar, 2002). This mechanism provides a typical set of seismic precursors which accompany the Merapi activities from magma ascent to eventual dome collapse (Ratdomopurbo and Poupinet, 2000). However, dome





collapse without any seismic precursors have occurred at Merapi before, as in November 1994, causing 60 fatalities and many more casualties (Voight et al., 2000b). Therefore, assessing the potential collapse of the Merapi lava dome during quiescence periods is vital for hazard mitigation.

In our model, the chosen source volume of $\sim 0.3 \times 10^6$ m$^3$ could produce a pyroclastic block and ash flows with a maximum

runout distance that exceeds 4 km. This runout distance is comparable to that of an eruption with a volcano explosivity index VEI=1. The dome collapse in 2006 involved a volume of $1 \times 10^6$ m$^3$, and travelled downhill along a distance of 4 km, destroyed the village of Kaliadem (Charbonnier and Gertisser, 2009;Ratdomopurbo et al., 2013). Same authors used Titan2D models to explain these block and ash flows, showing the reliability of this method for assessing future scenarios and providing us with important parameters to realize the forward simulation. According to our model simulation, no populated

area is currently directly located in the path of the flow that may result from a potential collapse of the southern dome sector. This is mainly because the affected area was previously already devastated by the 2010 eruption (Surono et al., 2012;Jenkins et al., 2013).

## 5. Conclusion

A detailed geomorphological study of the active Merapi lava dome reveals that the southern dome sector is potentially unstable. We used high resolution digital elevation models derived from TLS and camera drones to generate a high resolution topographic and photomosaic dataset. We map the morphology and structures and identified a 165 m long horseshoe-shaped fracture encircling the unstable dome sector. Thermal infrared images allow identifying fumaroles that follow the location of the main horseshoe-shaped fracture, obviously providing a dominant hydrothermal fluids pathway.

The structurally encircled sector has estimated volume of $\sim 0.3 \times 10^6$ m$^3$. While magma recharge, earthquakes and progressive alteration can all destabilize the sector, our factor of safety analysis indicates that even a typical rainfall event could increase the hydrothermal alteration activity and further destabilize the sector to a point where it may collapse. By using Titan2D flow simulation we estimate that collapse of the unstable dome sector may produce block and ash flows travelling southwardly with maximum run out distance of over 4 km from the summit.

**Acknowledgments**

This is a contribution to VOLCAPSE, which is a research project funded by the European Research Council under the European Union's H2020 Programme/ERC consolidator grant ERC-CoG 646858 and the authors also acknowledge a scholarship grant from Deutscher Akademischer Austauschdienst (DAAD), Germany reference number 91525854, a

research grant by the Swedish Research Council (VOLTAGE project), and financial support by the Swedish Center for Hazard and Disaster Sciences (CNDS). We would like to thank to Mehdi Nikkhoo, Nicole Richter for the data acquisition of Terrestrial Laser Scan, Michele Pantaleo for the data acquisition of thermal images in 2014, Julia Kubanek for providing the TanDEM-X dataset, Edgar Zon for supporting the 2014 fieldwork. We also thank to François Beauducel for supporting FLUKE camera during field work in 2017 and BPPTKG (Merapi Volcano Observatory) for all supports during field works





in 2014, 2015, and 2017. The high resolution DEM and photomosaic in this study are available in GFZ data publication with

DoI: http://doi.org/10.5880/GFZ.2.1.2017.003

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





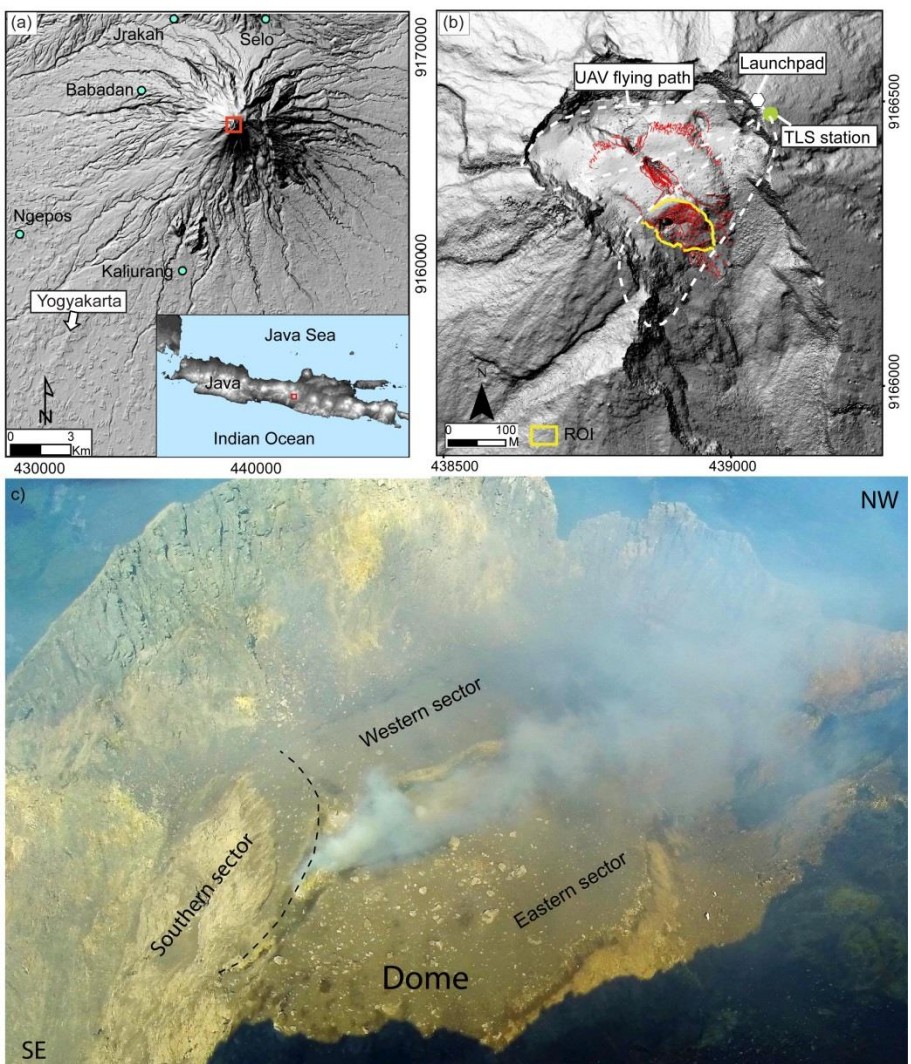

Figure 1. (a) Shaded relief of DEM from Gerstenecker et al. (2005) shows the morphology of Merapi Volcano, the most
active volcano in Indonesia. Merapi is located ~30 km from the densely populated city of Yogyakarta and therefore the
activity of Merapi is intensively monitored by five observatories (blue dots). (b) TLS and drone photogrammetry field
campaigns have been conducted in September 2014 and October 2015, respectively to investigate the detailed structure and
morphology of the Merapi lava dome. Coordinates are in UTM meters. (c) The aerial image of Merapi dome in 2014 shows
a delineated unstable dome sector at the southern flank of the dome that is the focus of the present investigation.





Figure 2. (a) Shaded relief map of the Merapi lava dome shows an elongated open fissure and a horseshoe-shaped structure that delineates an unstable dome sector at the southern part of the Merapi lava dome. (b) The delineated sector is located on a steep slope and it may fail and produces a pyroclastic avalanche. (c) Cross section profiles of lines p-q and (d) r-s show that the depth of the fractures along the horseshoe-shaped structure is between 3 and 8 meters deep at present. Coordinates are in UTM.



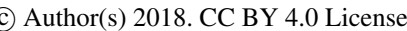



Figure 3. a) Photomosaic of UAV aerial images acquired in 2015 shows detail structures at the southern sector of the Merapi dome, (b) the lithology contact and we find three fractures zone (c, d, and e) at the horseshoe shaped structure. Coordinates are in UTM. Comparison of aerial images acquired in 2015 and 2017 reveals increasing of hydrothermal alteration intensity, localized at fractures in area c-c', d-d', and e-e' during the last 2-3 years. It strongly indicates a structural weakening occurs during the last 2-3 years up to now. We find five fractures in those three fractures area with diameter of the 1st, 2nd, 3th, 4th, and 5th fractures is 0.7 m, 0.3 m, 1 m, 1.3 m, and 0.3 m, respectively.





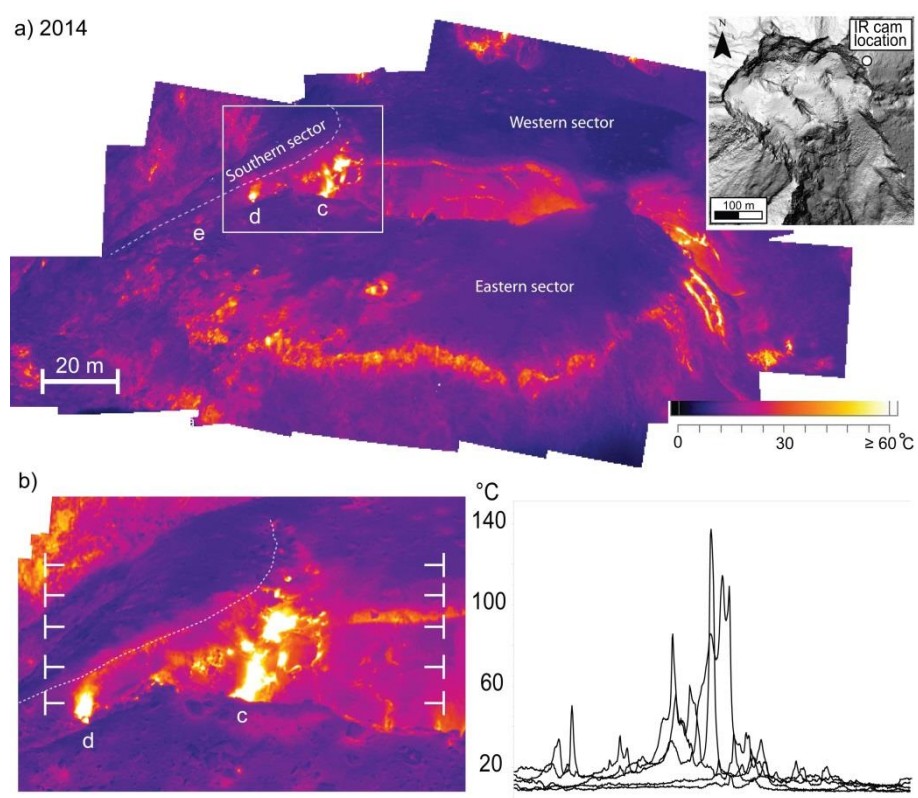

Figure 4. (a) Photomosaic of high resolution thermal image that taken from the eastern flank (inset) shows the thermal variation of the Merapi dome in 2014. (b) High temperature area strongly expressed in the southern dome sector, localized at the horseshoe shaped structure. Temperature profiles around this area show that the horseshoe-shaped structure has temperature between 20 - ≥140 °C.





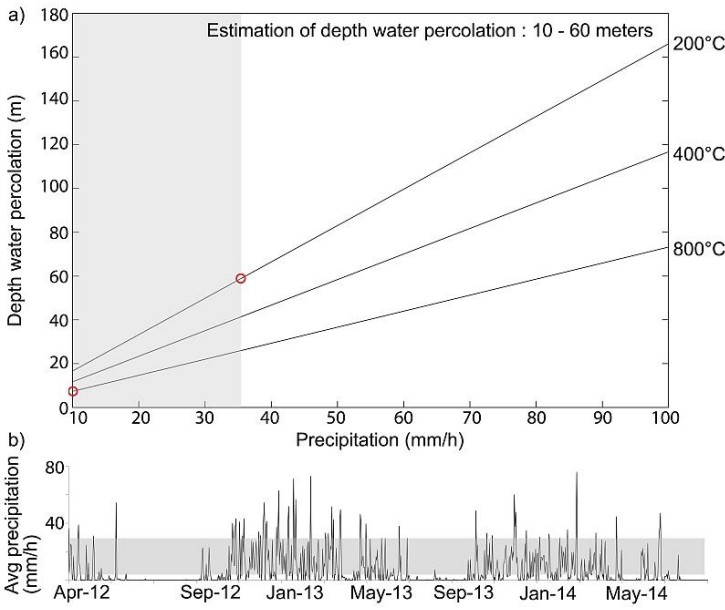

Figure 5 (a). Depth rain water percolation as a function of rainfall intensity in Merapi is controlled by the temperature of the dome. By assuming the minimum and maximum temperature of the dome of 200 and 800 °C, respectively, the estimation of depth water percolation is 10 – 60 meters (red circles) during typical rainfall in Merapi (grey area). (b) The typical intensity of rainfall from April 2012 to July 2014 was 10 – 35 mm/h (grey area) and it was calculated based on average of rain intensity from five observatories in Merapi (see Fig. 1a).





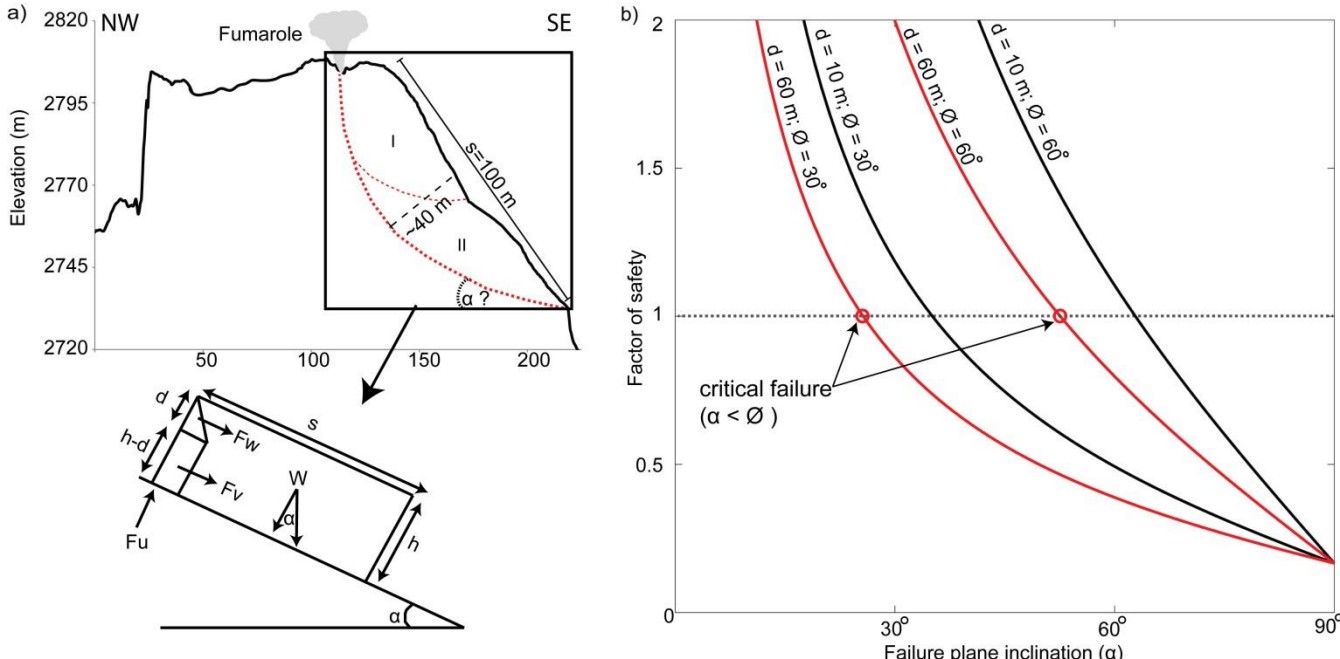

Figure 6. (a) Interpretative cross section of the unstable dome sector (Modified after Simmons et al. (2004)) illustrating that the horseshoe-shaped structure may develop a rotational listric fault with fracture spacing (s) up to 100 m and thickness of ~40 m. The instability of the unstable dome sector is influenced by the weight (W), water force ($F_W$), vaporized water force ($F_V$), and gas uplift force ($F_u$) along the fault boundary during intense rainfall. (b) Analysis of factor of safety for the entire sector shows that dome instability may occur when the inclined failure plane becomes steeper than the friction angle (α > Ø) in the shallow water percolation scenario (d = 10 m). For the deep water percolation scenario (d = 60 m, red lines), friction may not represent the main control any longer and instability may occur even with gentler slope angle (α < θ), since the factor of safety ≤ 1 indicates instability. We find that the critical failure inclination is 24° and 51° (red circles) for friction angle 30° and 60° in the deep water percolation scenario, respectively.



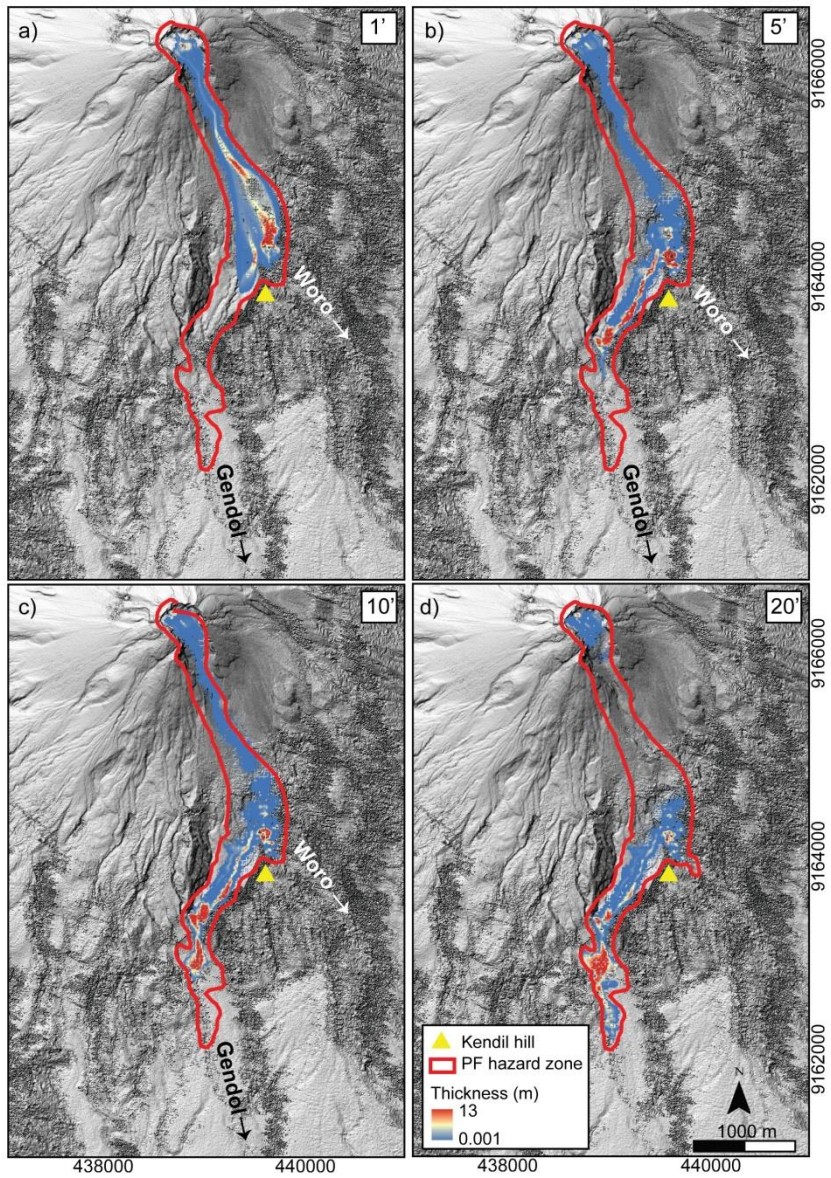

Figure 7. Result of pyroclastic block and ash flows due to collapse of the delineated unstable dome sector with volume of $0.3\times10^6$ m$^3$. Event duration of slices at 1, 5, 10, and 20 minutes are shown (deposited material is shown by blue and red areas). The block and ash flows are deflected by the Kendil hills (yellow triangle) within the first 5 minutes after the collapse. The red outline indicates the total block and ash flows hazard zone. However, Titan2D is limited in that it cannot simulate pyroclastic surges. Surges could potentially jump across the Kendil hill when the block and ash flows are forced to change direction within the Gendol valley. Coordinates are in UTM meters.





Table 1. Detail parameters to calculate depth water percolation (eq.1) and factor of safety (eq. 2)

| Parameters | Value | Source |
|---|---|---|
| Thermal diffusivity ($K_R$) | $1.4 \times 10^{-6}$ m²/s | Taron et al. (2007) |
| Heat capacity of rock ($c_R$) | 918 J/kg K | Taron et al. (2007); Simmons et al. (2004) |
| Heat capacity of water ($c_W$) | 4187 J/kg K | Taron et al. (2007); Simmons et al. (2004) |
| Rain duration ($t$) | 3 hours | Assumption |
| Rain intensity ($i$) | 10 – 100 mm/h | Data observation |
| Thermal to cool fracture ($\Delta T_R$) | 200 – 800° C | Thermal datasets and from Matthews and Barclay (2004). |
| Thermal to vaporized water ($\Delta T_W$) | 100°C | Assumption |
| Fracture spacing (s) | 100 m | Digital Elevation Model |
| Dome sector thickness (h) | ~40 m | Estimation |
| Density of rock ($\rho_r$) | 2242 kg/m³ | Tiede et al. [2005] |
| Density of water ($\rho_w$) | 1000 kg/m³ | Taron et al. (2007); Simmons et al. (2004) |
| Cohesive strength (Cs) | 10 MPa | Mayer et al. (2014); Pola et al. (2014) |
| Friction angle (Ø) | 30° - 60° | Barton and Choubey (1977); Stinton et al. (2004) |
| Gravitational acceleration ($g$) | 9.8 m²/s | |





Table 2. Detail input parameters in Titan2D simulation

| Parameters | Input data | Source |
|---|---|---|
| Topography model | Updated DEM | Drone photogrammetry + lidar+ TanDEM-X from *Kubanek et al.* [2015] |
| Number of flux source | 1 | Assumption |
| Duration (s) | 1200 s | Maximum time computation |
| Volume | 300.000 m$^3$ | DEM + failure plane inclination from FS analysis |
| Initial velocity | 0 | Assumption |
| Internal friction angle | 30° | Charbonnier et al. (2013) |
| Bed friction angle<br>Zone 1 : > 2426<br>Zone 2 : 2053 – 2425<br>Zone 3 : 1680 – 2052<br>Zone 4 : 1555 – 1679<br>Zone 5 : 1431 – 1554<br>Zone 6 : 1306 – 1430<br>Zone 7 : 1182 – 1305<br>Zone 8 : 0 - 1181 | <br>22°<br>20°<br>18°<br>16°<br>14°<br>12°<br>10°<br>8° | Charbonnier et al. (2013)<br>Zone 1 : > 2426<br>Zone 2 : 2053 – 2425<br>Zone 3 : 1680 – 2052<br>Zone 4 : 1555 – 1679<br>Zone 5 : 1431 – 1554<br>Zone 6 : 1306 – 1430<br>Zone 7 : 1182 – 1305<br>Zone 8 : 0 - 1181 |