# Peer review of "Structural weakening of the Merapi dome identified by drone photogrammetry after the 2010 eruption"

_Natural Hazards and Earth System Sciences, 2018_

## Short Comment (SC1) · 15 May 2018

The submitted manuscript provides a useful integrated study of drone-based geomorphological analysis and thermal infrared data collection to assess the stability of the dome of Merapi volcano. Water percolation within the dome is taken into consideration as trigger of dome collapses. The effort to provide a Factor of Safety is commendable. Although pyroclastic flow modelling is only a small portion of the research work illustrated here, to prevent this paper from being misleading, the authors should acknowledge the fact that there is still a lot of work to do before it is really possible to predict the mobility of pyroclastic flows.

[Figure]

I have a few important comments:

1) There is the need to mention the actual basal friction that the authors have chosen when running Titan2D: Coulomb, Voellmy or Pouliquen-Forterre, for example. If this is not done, it would be impossible to fully characterize the simulations.

2) Please recognize in the text that Titan2D, as the name confirms, is a two-dimensional model whose results are adapted to a three-dimensional subsurface only later on by the software package.

3) It is also very important to disclose that, in Titan2D, the flows never stop and the computer operator has to introduce an arbitrary criterion to decide when the flows cease their motion and a deposit is formed [Ogburn and Calder, 2017]. The lack of acknowledgment of this shortcoming generates the false notion that the pyroclastic flow mechanics is understood.

4) The main problem with Titan2D is that it ignores completely the granular nature of pyroclastic flows. This is in contrast to the fact that block-and-ash flows are well documented worldwide to be dense granular flows of angular rock fragments [Nairn and Self, 1978; Saucedo et al., 2002]. It is therefore important to inform the readers that an effort is undertaken to understand how rock fragments dissipate energy when interacting among themselves and the subsurface within travelling flows [e.g., Cagnoli and Piersanti, 2015 and 2017]. Since the grain size strongly affects the mobility, it is important to state clearly the grain size of the simulated flows.

5) My previous comments boil down to two questions. Considering that, block-and-ash flows are controlled by gravity and topography, do you really need Titan2D to know: A) that dome collapses discharge their rock debris down the deep and narrow valley which the horseshoe-shaped crater morphed into and B) that deposits form at the base of the volcanic cone where a dramatic change of the slope angle occurs?

Bruno Cagnoli, INGV

References

Cagnoli B., Piersanti A., 2015. Grain size and flow volume effects on granular flow mobility in numerical simulations: 3-D discrete element modeling of flows of angular rock fragments. J. Geophys. Res. Solid Earth, 120, 2350-2366, doi:10-1002/2014JB011729

Cagnoli B., Piersanti A., 2017. Combined effects of grain size, flow volume and channel width on geophysical flow mobility: three-dimensional discrete element modeling of dry and dense flows of angular rock fragments. Solid Earth, 8, 177-188.

Nairn I.A., Self S., 1978. Explosive eruption and pyroclastic avalanches from Ngauruhoe in February 1975. J. Volcanol. Geotherm. Res. 3, 39-60.

Ogburn S.E, Calder E.S., 2017. The relative effectiveness of empirical and physical models for simulating the dense undercurrent of pyroclastic flows under different emplacement conditions. Front. Earth Sci. 5:83, doi:10.3389/feart.2017.00083

Saucedo R., Macías J.L., Bursik M.I., Mora J.C., Gavilanes J.C., Cortes A., 2002. Emplacement of proclastic flows during the 1998-1999 eruption of Volcán de Colima, México. J. Volcanol. Geotherm. Res. 117, 129-153.

---

## Referee Comment (RC1) · Anonymous Referee #1 · 5 Jun 2018

Review of the manuscript: Dome instability at Merapi volcano identified by drone photogrammetry and numerical modeling by Herlan Darmawan, Thomas R. Walter, Valentin R. Troll, Agus Budi-Santoso

Dear Editor,

The paper of Darmawan et al. focuses on the potential destabilization of the frozen dome of Merapi volcano by rain. It first calculates the morphology of the summit dome of Merapi volcano, then the dome stability and, finally, the pyroclastic flows that can result from a collapse. My review follows the structure of the manuscript: title, morphology calculation, stability and pyroclastic flow modelling.

Title:

the title must be changed to indicate that the studies focuses on the effect of rain and is not a comprehensive study of all the mechanisms than can lead to the collapse of active domes, as suggested by the current title.

Morphology:

The data obtained from Terrestrial Laser Scanning (TLS) and from drone are impressive and shows the power of such methods for calculating the morphology of dangerous areas. For the topography, most of the data were already published and the novelty here, is to extract the topography profile as well as the fumarole locations from visible images. Another novelty is to couple thermal images to confirm the fumaroles locations. My only doubt on this part is the temperatures accuracy given by the authors. According to variation of the atmosphere humidity, the composition of magmatic gas and the variable distance from the camera (it seems that a mean distance of 300 m is taken into account for the correction instead of the real distance, calculated with the DEM) and the pixel size, an error of the temperature of only 3°C seems very accurate. Could the authors give more details on how they have obtained this accuracy estimation? If not, it can be stated that the temperature is approximate, which is enough for the needs of the manuscript. Line 29 of section 3.1 must also be modified: as fumaroles activity is also related to rain, a punctual observation of more visible fumaroles can be related to seasonal changes and not necessarily to an increase of the activity.

Factor of safety:

This section is essentially based on the work and the model of Simmons et al. (2004). The novelty is the application to Merapi. My main criticism is that it is not easy to understand the calculations that have been done, and that some formula are perhaps wrong.

First, it must be explained why the authors focus on a small portion of a frozen dome.

All the summit, including the frozen dome, is cut by fractures. The crater flanks are very steep and can also collapse. The whole summit must be studied for a complete study of destabilizations.

Even if the safety model has been developed by others, the reader needs some information to understand what has been done (even briefly). Among the questions: what are the basis of the model? Why is there a link between the depth of water percolation and the distance between fractures to the square? (this is probably related to the surface that supplies the fracture, but why to the square?). Why the fracture widths are not taken into account in the percolation depth calculation? The "forces" must be more clearly explained and I recommend to expand and to detail the scheme of Fig. 6. The formulation of Fw and Fv are correct but it needs explanations: why a coefficient 0.5? Explain why, to calculate the force of the volcanic gas Fv, the density of the liquid and not of the gas is used (I have understood only by reading related papers). The "forces" W and F are not real forces (in N) but forces per meter in (N/m). It might be called a force but after being defined correctly.

In Equ. 2, C must be a force per meter. Is it the same as Cs, in tab. 1, both called "cohesive strength" but with a unit of stress (Pa)? The authors used the formulation of Simmons et al. and reproduce a probable typo in the formula (Eq. 1 of Simmons et al., 2004): Cs was probably C*s (in N/m in this case). Are the results obtained with a correct formula or with C instead of C*s? Because s = 100 m, using C instead C*s will significantly change the results. I cannot understand what is Fu and how it is calculated. If it is the water pressure at the base of the dome, the "force" must equal the pressure at the base of the fracture multiply by the dome surface, and it must be: Fu = d*cos(a)*rhow*g*s (neglecting the gas density). Where does the coefficient 0.5 come from? A progressive pressure decrease to the front? Why is it called the "uplift force from the volcanic gas", if it is related to the pressure of the liquid water only?

Once all these points will be fixed / clarified, the other point is the sensitivity of the model. The authors say that the "calculation requires careful parameter justification"

[Figure]

(section 5). As several parameters seem estimated roughly, other graphs like that of Fig. 6 are needed to explore the model sensitivity to the cohesive strength, the temperature, the volume rate of the rain, the fracture spacing, etc on the stability. I think that friction angles of $60°$ are not realistic and it can be replaced by a friction angle of $20°$ and $40°$.

Modelling of pyroclastic flows:

The volume that can collapse in this manuscript is small and the lava dome is cold for several years (except because it transmits the temperature of the gases). In this case, why do the authors expect the genesis of pyroclastic flows? Even pyroclastic surges are evoked (4.1, p. 11, line 1-2). This assumption seems surprising and needs to be explained clearly. Works cited (e.g. Elsworth et al.) focus on an active and hot lava domes. The limitation of the numerical model used must also be presented. For example, the deposits of figure 7 are not compatible with pyroclastic flow deposits. They accumulate at the foot of the volcano forming piles more compatible with small rock collapse. Because the volume is small and the dome is cold, it is probable that a collapse will form a rock avalanche and not a pyroclastic flow but this must be explained and parametrized clearly.

If the authors want to simulate pyroclastic flows, a long debate exists about the models and the approaches used for pyroclastic flows and, today, models of pyroclastic flows are not reliable enough to be presented without discussions and caution. In this context, two points seem very worrying to me: 1) the work recently published by Kelfoun et al (2017) on the same theme (numerical simulation of pyroclastic flows) and on the same volcano (Merapi, 2010) is neither cited nor discussed. It cannot be ignored even if the model seems to reproduce correctly a pyroclastic flow emplacement with a physics that differs from the physics of the present manuscript. 2) the references to Charbonnier et al (2013) are partial. Their work is cited to justify that Titan2D is a tool that makes good simulations of pyroclastic flows, avoiding discussions on the model limitations. However, even if they have shown positive features, Charbonnier et

al. have also shown the limitations of the models. For example, they wrote: "Titan2D is not capable of reproducing the runout distances and areas covered by the actual events over the highly complex topography" (discussion, 5.3.2). Is it compatible with its use in the present manuscript? A model is never perfect and it is why the limitations of the approach and of the results must be clearly and honestly discussed. The remarks of SC1 on the interactive discussion are also significant: for example, is the simulation able to stop? If not, what criterion has been chosen?

The quality of the DEM used, which seems to be very noisy, and the consequences on the results needs to be discussed too. Finally, given all the limitations of the approach and because the shape of the volcano has not changed from the last eruption, I wondered something similar to SC1: does the numerical model presented give results more confident than a rough estimation based on the past experience of Merapi's eruptions?

My conclusion is that, even if the data are interesting, they have been already partially published. The calculation of the stability is not new (except that it is applied to Merapi), not detailed enough and, maybe, partially wrong (C/Cs and Fu). The study is focussed on a very local problem: the collapse of a small part of a frozen lava dome by rain. The simulation of pyroclastic flows is based on a questionable assumption (a cold lava dome can create pyroclastic flow) and, the limitations of the model used and the results are not detailed enough. In the current state, I think that the paper cannot be published and it must be deeply reworked before publication.
* * *

---

## Referee Comment (RC2) · S. Charbonnier (Referee) · 12 Jun 2018

This paper investigates dome instability at Merapi after the 2010 eruption. Although I agree that this kind of study is of primary importance to better assess the related hazards associated with the future occurrence of dome-collapse events at Merapi, I would only recommend this manuscript for publication in NHESS journal after major revisions.

Firstly, the language used by the authors in the text is sometimes limited and confusing. Many paragraphs are not readable and/or comprehensible, too long and repetitive, and some even lack of meaning. I have outlined some of the main issues in the attached

[Figure]

PDF but could not pay attention to every typo, grammar, repetitions, waffles and sentence structure problems. I would invite the authors to entirely revise some parts of the manuscript by taking into account the comments included in the attached PDF. Secondly, the scientific part of the manuscript is somehow incomplete in some aspects. Although some of the issues related to dome stability are correctly described and discussed, some of the concepts presented in this paper lack of new innovative ideas. The authors should rather focused on the recent structural features that developed in the entire summit area, including the crater rim and upper part of the cone, and not only the post-2010 lava dome. I think the recent 2018 explosive events should be taken into account, especially for the results presented in figure 5 and 6 showing the link between water percolation and slope failure as well as the deep structure of the summit area; but also for the discussion about flow hazard assessment, given the high potential of larger hazard associated with a larger scale event! The authors also completely misunderstood the use of varying basal friction angles associated with different flow volumes, as explained in details in Charbonnier and Gertisser (2012). I suggest them to read carefully the paper and change the basal friction angles accordingly. An explanation about why Titan2D cannot model surges is also lacking...

Finally, the discussion section is badly written and should focus more about the results shown in section 3, particularly the structural and geomorphological data obtained, rather than just conversing on dome collapse hazards at Merapi. This could considerably straighten some of the interesting results obtained in this study by justifying the use of some new innovative techniques (TLS, SfM) to solve the issues outlined in the previous sections.

Please also note the supplement to this comment:
https://www.nat-hazards-earth-syst-sci-discuss.net/nhess-2018-120/nhess-2018-120-RC2-supplement.pdf
* * *
[Figure]

2018-120, 2018.

**Supplement:**

[revised manuscript text omitted]

Zone 1 : > 2426
Zone 2 : 2053 – 2425
Zone 3 : 1680 – 2052
Zone 4 : 1555 – 1679
Zone 5 : 1431 – 1554
Zone 6 : 1306 – 1430
Zone 7 : 1182 – 1305
Zone 8 : 0 - 1181 |
22°
20°
18°
16°
14°
12°
10°
8° | Charbonnier et al. (2013)
Zone 1 : > 2426
Zone 2 : 2053 – 2425
Zone 3 : 1680 – 2052
Zone 4 : 1555 – 1679
Zone 5 : 1431 – 1554
Zone 6 : 1306 – 1430
Zone 7 : 1182 – 1305
Zone 8 : 0 - 1181 |

---

## Author Comment (AC1) · 8 Aug 2018

SC: The submitted manuscript provides a useful integrated study of drone-based geomorphological analysis and thermal infrared data collection to assess the stability of the dome of Merapi volcano. Water percolation within the dome is taken into consideration as trigger of dome collapses. The effort to provide a Factor of Safety is commendable. Although pyroclastic flow modelling is only a small portion of the research work illustrated here, to prevent this paper from being misleading, the authors should acknowledge the fact that there is still a lot of work to do before it is really possible to predict the mobility of pyroclastic flows.

Response: Thank you very much for the comments. We agree to the points that the modelling is only a small portion of the work, and that there is still a lot of work to do before PDC can be predicted, and that especially the modelling technique used is limited. Therefore we are more careful with the interpretation of our result and inserted a critical discussion.

SC: I have a few important comments: 1) There is the need to mention the actual basal friction that the authors have chosen when running Titan2D: Coulomb, Voellmy or Pouliquen-Forterre, for example. If this is not done, it would be impossible to fully characterize the simulations.

Response: comment accepted. Titan2D uses the Coulomb friction to simulate geo-physical mass flow over natural terrain. We have now added this information in the revised version.

2) Please recognize in the text that Titan2D, as the name confirms, is a two-dimensional model whose results are adapted to a three-dimensional subsurface only later on by the software package.

Response: Accepted. We added the 2-D limitation in the introduction and in the method sections, where we describe the application and basic theory of Titan2D.

3) It is also very important to disclose that, in Titan2D, the flows never stop and the computer operator has to introduce an arbitrary criterion to decide when the flows cease their motion and a deposit is formed [Ogburn and Calder, 2017]. The lack of acknowledgment of this shortcoming generates the false notion that the pyroclastic flow mechanics is understood.

Response: comment accepted, it is true that Titan2D will not technically stop in the end of simulation. This we clarified in the revised version. As we note, the velocity of the flow will exponentially decrease to $\sim$0 m/s when the computation reached maximum time simulation (Charbonnier and Gertisser, 2009). In order to obtain more realistic

rock avalanche model, we set validated coulomb friction angle and changed the maximum simulation time from 20 minutes to 1 hour, being aware that the accurate timing must not misinterpreted with true timing. We have added this clarification in the revised version.

4) The main problem with Titan2D is that it ignores completely the granular nature of pyroclastic flows. This is in contrast to the fact that block-and-ash flows are well documented worldwide to be dense granular flows of angular rock fragments [Nairn and Self, 1978; Saucedo et al., 2002]. It is therefore important to inform the readers that an effort is undertaken to understand how rock fragments dissipate energy when interacting among themselves and the subsurface within travelling flows [e.g., Cagnoli and Piersanti, 2015 and 2017]. Since the grain size strongly affects the mobility, it is important to state clearly the grain size of the simulated flows.

Response: We appreciate this comment. Titan2D software not completely ignores the granular nature of pyroclastic flows/debris avalanches. The flows are assumed to be incompressible continuum and the interaction between grains-grains and grains-basal surface is solved by Mohr-Coulomb law (see Patra et al., 2005). To discuss the limitations of the models and efforts of studying rock fragments in granular flows, we inserted the suggested references (Cagnoli and Piersanti, 2015 and 2017). We better clarify the momentum effects due to grain size interaction. We added further details of the basic theory of Titan2D in the methods section and more thoroughly discuss the limitation in the revised version.

5) My previous comments boil down to two questions. Considering that, block-and-ash flows are controlled by gravity and topography, do you really need Titan2D to know: A) that dome collapses discharge their rock debris down the deep and narrow valley which the horseshoe-shaped crater morphed into and B) that deposits form at the base of the volcanic cone where a dramatic change of the slope angle occurs?

Response: We appreciate this comment. Yes, it is necessary to define source collapse

**NHESSD**

mechanism in Titan2D. Titan2D is able to model several collapse scenarios such as a single collapse, multiple collapses, a gravitational collapse, or a fountain collapse that produce radial debris avalanches. In our model, we define that the mechanism of the source collapse is a single block collapse which triggered by hydrothermal alteration and neglect gas overpressure. Therefore, we chose a single gravitational collapse scenario (flux model) and set initial velocity of 0 m/s (no gas overpressure) and volume of 500.000 m3 (volume of delineated block). The deposit and the flow mechanism of Titan2D simulation are controlled by coulomb friction. In order to obtain realistic model where the flow is controlled by topography and different slope, we applied material map, which integrated with DEM. We defined variation of coulomb friction based on slope variation. In order to clarify the mechanism of source collapse and the debris flow, we added detail description of parameters that control the source collapse and the variation of coulomb friction angles in the revised version.

---

## Author Comment (AC2) · 9 Aug 2018

Reviewer: This paper investigates dome instability at Merapi after the 2010 eruption. Although I agree that this kind of study is of primary importance to better assess the related hazards associated with the future occurrence of dome-collapse events at Merapi, I would only recommend this manuscript for publication in NHESS journal after major revisions. Firstly, the language used by the authors in the text is sometimes limited and confusing. Many paragraphs are not readable and/or comprehensible, too long and repetitive, and some even lack of meaning. I have outlined some of the main issues in the attached PDF but could not pay attention to every typo, grammar, repe-

titions, waffles and sentence structure problems. I would invite the authors to entirely revise some parts of the manuscript by taking into account the comments included in the attached PDF.

Response: We appreciate this comment and made appropriate changes. We have revised and deeply re-worked the editing and language. We checked and corrected the grammar mistakes, deleted some typos and repetitive sentences, and re-phrased or deleted confusing paragraphs in order to improve the manuscript. A native speaker was proofreading the manuscript and found further language deficiencies that could be corrected.

Reviewer: Secondly, the scientific part of the manuscript is somehow incomplete in some aspects. Although some of the issues related to dome stability are correctly described and discussed, some of the concepts presented in this paper lack of new innovative ideas.

Response: Accepted comment and changes made. We further clarified the novelties of the work. We conducted the first geomorphology, thermal and structural mapping of the southern dome at Merapi. We are able to identify sub meter fractures and quantify the structural pattern of the unbuttressed dome sector in detail. The fractures are actively degassing as identified by our thermal camera. The geomorphology, structure, and thermal datasets were then used to investigate a potential hazard using two methods, factor of safety and Titan2D. Application of the factor of safety calculation from Simmons et al (2004) and Titan2D (previously used at Merapi by Charbonnier and Gertisser (2009; 2012) we are able to evaluate the hazard arising from this unstable dome sector. We therefore think that the paper contains innovations justifying publication, which we could now further clarify in the revised version.

Reviewer: The authors should rather focused on the recent structural features that developed in the entire summit area, including the crater rim and upper part of the cone, and not only the post-2010 lava dome.

Response : Accepted comment and changes made. We re-analyzed the geomor-phology, structures, thermal distribution and alteration area at the summit of Merapi. We also re-calculated the factor of safety for the south and the west flanks, which are progressively altered and thus experience structural instability in the near future.

I think the recent 2018 explosive events should be taken into account, especially for the results presented in figure 5 and 6 showing the link between water percolation and slope failure as well as the deep structure of the summit area; but also for the discussion about flow hazard assessment, given the high potential of larger hazard associated with a larger scale event!

Response: comment accepted and changes made. We added a short discussion and relationship of the 2018 explosions into the discussion section. As the dome is also subjected to hydrothermal alteration, we assess and re-calculate the flanks instability by using factor of safety of Bishop's and Swedish equation, which was also suggested by second reviewer. We further explained parameters and forces that influence the factor of safety calculation more detail in the revised version. Discussion and limitation of each parameter is also added in the revised version.

The authors also completely misunderstood the use of varying basal friction angles associated with different flow volumes, as explained in details in Charbonnier and Ger-tisser (2012). I suggest them to read carefully the paper and change the basal friction angles accordingly. An explanation about why Titan2D cannot model surges is also lacking...

Response : Thank you very much for the suggestion. We agree that basal friction angles should be considered with care, and we understand the limitations. We have read carefully the paper and change the basal friction angle according to Charbonnier and Gertisser (2009 and 2012). We added the limitation of Titan2D parameters in the revised version. Concerning the comment on "surges", we agree that Titan2D cannot model this phenomenon.

Finally, the discussion section is badly written and should focus more about the results shown in section 3, particularly the structural and geomorphological data obtained, rather than just conversing on dome collapse hazards at Merapi. This could considerably straighten some of the interesting results obtained in this study by justifying the use of some new innovative techniques (TLS, SfM) to solve the issues outlined in the previous sections.

Response : Thank you for the critical comments. We thoroughly rewrote the discussion section. We added more detail about the geomorphology and structure at the summit. After the climactic eruption in 2010, the morphology of the Merapi summit has changed dramatically. As we describe in the revised version, previous study concerned the dramatic topographic changes (Kubanek et al., 2013) by comparing satellite radar data before and after the eruption and calculating the volume of the 2010 eruption. Our datasets now provides a much better resolution of the changes occurring within this newly developed crater, and could describe more realistic condition of the current morphology and structure of the Merapi summit. We now outline this aspect in our revised manuscript. In addition, geomorphology and structural mapping at a steep-sided dome building volcano such as Merapi is challenging. For this we have realized a method linking lidar and drone based mapping. Combination of TLS and SfM is very promising and provides high quality datasets to identify sub-meter fractures and slope changes. This is the first study to map geomorphology and structure at the Merapi summit by using these techniques. Advantageous and limitations toward these techniques are further discussed in the revised version. To further improve the flow and organization of the discussion section, we let it proofread by native speaking.

Please also note the supplement to this comment:
https://www.nat-hazards-earth-syst-sci-discuss.net/nhess-2018-120/nhess-2018-120-AC2-supplement.pdf

2018-120, 2018.

**Supplement:**

[revised manuscript text omitted]

| friction angle
Zone 1 : > 2426
Zone 2 : 2053 – 2425
Zone 3 : 1680 – 2052
Zone 4 : 1555 – 1679
Zone 5 : 1431 – 1554
Zone 6 : 1306 – 1430
Zone 7 : 1182 – 1305
Zone 8 : 0 - 1181 | 22°
20°
18°
16°
14°
12°
10°
8° | Charbonnier et al. (2013)
Zone 1 : > 2426
Zone 2 : 2053 – 2425
Zone 3 : 1680 – 2052
Zone 4 : 1555 – 1679
Zone 5 : 1431 – 1554
Zone 6 : 1306 – 1430
Zone 7 : 1182 – 1305
Zone 8 : 0 - 1181 |

---

## Author Comment (AC3) · 9 Aug 2018

Reviewer: Dear Editor, The paper of Darmawan et al. focuses on the potential destabilization of the frozen dome of Merapi volcano by rain. It first calculates the morphology of the summit dome of Merapi volcano, then the dome stability and, finally, the pyroclastic flows that can result from a collapse. My review follows the structure of the manuscript: title, morphology calculation, stability and pyroclastic flow modelling. Title: The title must be changed to indicate that the studies focuses on the effect of rain and is not a comprehensive study of all the mechanisms than can lead to the collapse of active domes, as suggested by the current title.

[Figure]

Response: We appreciate this comment and made appropriate changes. We have changed the title to: "Structural instability of the dome at Mt. Merapi volcano identified by drone photogrammetry and modeling."

Reviewer: Morphology: The data obtained from Terrestrial Laser Scanning (TLS) and from drone are impressive and shows the power of such methods for calculating the morphology of dangerous areas. For the topography, most of the data were already published and the novelty here is to extract the topography profile as well as the fumarole locations from visible images.

Response: Yes, part of the data is published earlier. In this new study we added new close up views of a photomosaic generated from new drone overflights. This new data adds further information about the location of fumaroles. Another very important new dataset is the use of high resolution thermal infrared maps. These were generated by a superzoom lens and image mosaicing. The results allow is identifying precise positions of gas escape. This gas escape follows a structural pattern already inferred from optical data, and weakly expressed in the Terrestrial Laser Scanning results. Therefore this paper presents a number of novel and innovative methods. In the revised version we improved this description and made the novelties further clear.

Reviewer: Another novelty is to couple thermal images to confirm the fumaroles locations. My only doubt on this part is the temperatures accuracy given by the authors. According to variation of the atmosphere humidity, the composition of magmatic gas and the variable distance from the camera (it seems that a mean distance of 300 m is taken into account for the correction instead of the real distance, calculated with the DEM) and the pixel size, an error of the temperature of only 3°C seems very accurate. Could the authors give more details on how they have obtained this accuracy estimation? If not, it can be stated that the temperature is approximate, which is enough for the needs of the manuscript. Line 29 of section 3.1 must also be modified: as fumaroles activity is also related to rain, a punctual observation of more visible fumaroles can be related to seasonal changes and not necessarily to an increase of the activity.

Response: Accepted comment and changes made. We follow the reviewers suggestion and describe the temperature as approximate, which is indeed enough for the needs of the manuscript. Nevertheless we assess the temperature uncertainty following (Spampinato et al., 2011). By varying parameters of emissivity, distance, reflection temperature (Trefl), atmospheric temperature (Tatm), relative humidity (RH), computed transmission, external optics temperature, and external optics transmission, we could assess the uncertainty. The uncertainty was obtained by choosing one pixel in the same area, varying one parameter, and then calculating the RMSE (see table 1 below). Based on the calculation, we found that increasing emissivity by 0.01 may influence the apparent temperature of 1.04°C. Other studies of the dome rock emissivity at volcanoes (Merapi, Carr et al., 2016) and Colima (Walter et al., 2013) suggested that the emissivity may be in the range of 0.95 and 0.98, therefore, we estimate that the uncertainty of the thermal pixel value is ~3°C. However, in order to improve the manuscript, we accepted the suggestion from the reviewer by clarifying that the temperature is approximate in the revised version.

Reviewer: Factor of safety: This section is essentially based on the work and the model of Simmons et al. (2004). The novelty is the application to Merapi. My main criticism is that it is not easy to understand the calculations that have been done, and that some formula are perhaps wrong.

Response: Accepted comment and changes made. Following language proofreading, some of the unclear phrasing might already improve the clarity of the text. Furthermore, we improve the description of the safety equation (FS). Factor of safety is widely used to calculate slope instability and it is calculated by dividing resisting forces to driving forces that acting on a failure plane ($\alpha$). The conventional model such as Slice, Swedish, Bishop's methods are commonly used to calculate slope instability. However, in an active lava dome, some additional forces may influence the resisting and driving forces. The FS equation used by us is based on Simmons's work, which aims to calculate dome instability during intense rainfall. Rain water may build up gas (Fu), vaporize the

water (Fv), and add water forces (Fw). The FS from Simmon et al. (2004) considers the uplift force that may reduce the resistance force (W.cos($\alpha$)) and the water and vaporized water forces that may add the driving force (W.sin($\alpha$)). Therefore, we do not think that the equation is wrong, but we improved the text flow. As the current Merapi lava dome is influenced by degassing and rainfall activities, we used the FS equation from Simmons et al. (2004) to estimate the failure plane inclination ($\alpha$), therefore, we were able to quantify the volume of source collapse. In order to make the equation more understandable, we clarified each parameter in the method (section 2.2) in the revised manuscript. We also re-calculate the factor of safety by using Swedish and Bishop's method to compare our FS results in the revised version.

Reviewer: It must be explained why the authors focus on a small portion of a frozen dome. All the summit, including the frozen dome, is cut by fractures. The crater flanks are very steep and can also collapse. The whole summit must be studied for a complete study of destabilizations.

Response: very good comments. Obviously our description of the horseshoe shaped fracture and the instability tests applied for a particular dome sector were not clear. We focus on a small portion of the Merapi dome because we find a structural weakening due to hydrothermal alteration at the southern part of the lava dome. This structural weakening is evidenced based on digital elevation models showing a horseshoe shaped crater, a fumarole expression following this horseshoe shaped pattern, and degassing of hot fluids along such a horseshoe shaped fluid pathway. It has been explained in the introduction that hydrothermal alteration may weaken dome rocks and promote dome collapse (L7-13, page 2). We improved the description of the horseshoe shaped fracture in the revised version. However, the idea to assess dome instability of the whole summit is good and accepted. We added instability analysis at the western flank in the revised manuscript as this area is also subjected by progressive hydrothermal alteration. Previous studies also suggest that the dome collapses were dominantly to the west-south west direction in 1900's (Voight et al., 2000).

Reviewer: Even if the safety model has been developed by others, the reader needs some information to understand what has been done (even briefly). Among the questions: what are the basis of the model?

Response: We introduced the factor of safety model in the revised version to describe the application of this model to the readers. As mentioned before that the factor of safety is generally used to assess slope stability. The model is calculated by comparing resisting force to driving force that acts on a failure plane. We added the basic concept of factor of safety in the revised version, compare our FS results from Simmons et al. (2004) to the FS results from Swedish and Bishop's model.

Reviewer: Why is there a link between the depth of water percolation and the distance between fractures to the square? (this is probably related to the surface that supplies the fracture, but why to the square?).

Response: The instability of the square is influenced by water percolation (d) and the fracture spacing (s). Sensitivity tests of these two parameters suggested that increasing fracture spacing slightly decreases the factor of safet, while increasing water percolation (d) three times may reduce the stability 0.16 to 0.27 (Simmons et al., 2005).

Reviewer: Why the fracture widths are not taken into account in the percolation depth calculation?

Response : a critical question, which also leads us to further improve the discussion section of the paper. The water percolation is calculated based on equation 1 that consider a fracture spacing (s) on the block and dome properties and neglect the fracture width parameter. The equation considers that the dome properties (temperature, heat capacity, and thermal diffusivity) have stronger control toward water infiltration than the fracture width (Fig. 6). Therefore fracture width is not necessarily to be taken into account in the percolation depth calculation.

Reviewer: The "forces" must be more clearly explained and I recommend to expand

and to detail the scheme of Fig. 6. The formulation of Fw and Fv are correct but it needs explanations: why a coefficient 0.5? Explain why, to calculate the force of the volcanic gas Fv, the density of the liquid and not of the gas is used (I have understood only by reading related papers).

Response : We added detail explanation of forces that influence the FS calculations in the revised version. Fw and Fv are water and vaporized water forces, respectively. The coefficient of 0.5 is to calculate the geometry volume of Fw that influence the block (see Fig. 6). In general, force can be expressed by multiplying mass and acceleration/gravity acceleration ($F = m \cdot g$, where $m = v \times$, so $F = v \times \times g$ ). In the equation, it is mentioned that:

$Fw=(0.5 \times d2 \times \cos(\alpha)) \times w \times g$.

We inferred that the 0.5 is a constant that used to calculate the geometry volume of Fw that influence the block stability (see Fig. 6 where Fw are represented as triangle prism object). We used the density of water to calculate Fv as Fv represents the force of vaporized water from rainfall that interacts with hot dome interior.

Reviewer: The "forces" W and F are not real forces (in N) but forces per meter in (N/m). It might be called a force but after being defined correctly.

Response : accepted comments. We corrected, converted all parameters in SI units and re-calculated the Factor of Safety in the revised version.

Reviewer: In Equ. 2, C must be a force per meter. Is it the same as Cs, in tab. 1, both called "cohesive strength" but with a unit of stress (Pa)? The authors used the formulation of Simmons et al. and reproduce a probable typo in the formula (Eq. 1 of Simmons et al., 2004): Cs was probably C*s (in N/m in this case). Are the results obtained with a correct formula or with C instead of C*s? Because s = 100 m, using C instead C*s will significantly change the results.

Response : Accepted comments. We corrected Cs to C*s, converted all parameters in

SI units, re-calculated the FS from Simmons et al.(2004) and compared the FS results from Simmons et al to the FS calculation based on Swedish and Bishop's methods.

Reviewer: I cannot understand what is Fu and how it is calculated. If it is the water pressure at the base of the dome, the "force" must equal the pressure at the base of the fracture multiply by the dome surface, and it must be: Fu = d*cos(a)*rhow*g*s (neglecting the gas density). Where does the coefficient 0.5 come from? A progressive pressure decrease to the front? Why is it called the "uplift force from the volcanic gas", if it is related to the pressure of the liquid water only?

Response : The uplift force (Fu) is produced by water vapor and volcanic gas that released upward through the fracture to atmosphere. As mentioned before that the coefficient 0.5 is a constant that used to calculate the volume of Fu that influence the block stability (see Fig. 6 where Fu are illustrated by triangle prism object). Fu=(0.5×d2×cos($\alpha$)×s)×w×g according to Simmons et al (2005)

Reviewer: Once all these points will be fixed / clarified, the other point is the sensitivity of the model. The authors say that the "calculation requires careful parameter justification" (section 5). As several parameters seem estimated roughly, other graphs like that of Fig. 6 are needed to explore the model sensitivity to the cohesive strength, the temperature, the volume rate of the rain, the fracture spacing, etc on the stability. I think that friction angles of 60_ are not realistic and it can be replaced by a friction angle of 20° and 40°.

Response : accepted comments and suggestions. In the revised version, we clearly described the limitation of the FS method and the sensitivity of the parameters. We recalculated the factor of safety by using friction angle of 25° and 45° according studies from Simmons et al (2005) and Husein et al (2014).

Modelling of pyroclastic flows:

Reviewer: The volume that can collapse in this manuscript is small and the lava dome

is cold for several years (except because it transmits the temperature of the gases). In this case, why do the authors expect the genesis of pyroclastic flows? Even pyroclastic surges are evoked (4.1, p. 11, line 1-2). This assumption seems surprising and needs to be explained clearly. Works cited (e.g. Elsworth et al.) focus on an active and hot lava domes. The limitation of the numerical model used must also be presented. For example, the deposits of figure 7 are not compatible with pyroclastic flow deposits. They accumulate at the foot of the volcano forming piles more compatible with small rock collapse. Because the volume is small and the dome is cold, it is probable that a collapse will form a rock avalanche and not a pyroclastic flow but this must be explained and parametrized clearly.

Response : Accepted comments. We changed the terminology from pyroclastic flows to rock avalanche as the collapse of southern frozen dome is likely to produce rock avalanche than pyroclastic flow. However, we think that it is vital to assess the potential hazard zone in a case of the southern dome collapse as many cases suggested that hydrothermal alteration may weaken the rock and trigger a collapse. In addition, sand mining intensively occurs at the southern flank with radius of 5 km from the summit of Merapi. This is the reason why we assess the potential hazard of the southern Merapi dome sector. The idea to assess the potential hazard in this manuscript is vital for hazard assessment in the future.

Reviewer: If the authors want to simulate pyroclastic flows, a long debate exists about the models and the approaches used for pyroclastic flows and, today, models of pyroclastic flows are not reliable enough to be presented without discussions and caution. In this context, two points seem very worrying to me: 1) the work recently published by Kelfoun et al (2017) on the same theme (numerical simulation of pyroclastic flows) and on the same volcano (Merapi, 2010) is neither cited nor discussed. It cannot be ignored even if the model seems to reproduce correctly a pyroclastic flow emplacement with a physics that differs from the physics of the present manuscript. 2) the references to Charbonnier et al (2013) are partial. Their work is cited to justify that Titan2D is

a tool that makes good simulations of pyroclastic flows, avoiding discussions on the model limitations. However, even if they have shown positive features, Charbonnier et al. have also shown the limitations of the models. For example, they wrote: "Titan2D is not capable of reproducing the runout distances and areas covered by the actual events over the highly complex topography" (discussion, 5.3.2). Is it compatible with its use in the present manuscript? A model is never perfect and it is why the limitations of the approach and of the results must be clearly and honestly discussed. The remarks of SC1 on the interactive discussion are also significant: for example, is the simulation able to stop? If not, what criterion has been chosen?

Response : Accepted comments. We changed the terminology to rock avalanche as suggested in the previous point. We still used Titan2D to model the rock avalanche as titan2D is well-validated to model granular avalanches over natural terrain (Patra et al. (2005); Pitman et al. (2003). The suggestion to add the work from Kelfoun et al (2017) is accepted and added in the discussion. We agree that model is never perfect, therefore, we added more detail on limitation of the Titan2D to simulate debris avalanche in complex topography as also suggested by Charbonnier (second reviewer). The remark of SC1 is also correct that Titan2D has limitation to stop the simulation. The simulation cannot perfectly stop, even it is reached the maximum time simulation. In order to fit realistic model, we extended the maximum time simulation up to 1 hour which is long enough for rock avalanche duration and set validated/corrected friction angle parameter as this parameter control the run out and distribution of the rock avalanche.

Reviewer: The quality of the DEM used, which seems to be very noisy, and the consequences on the results needs to be discussed too. Finally, given all the limitations of the approach and because the shape of the volcano has not changed from the last eruption, I wondered something similar to SC1: does the numerical model presented give results more confident than a rough estimation based on the past experience of Merapi's eruptions?

Response : We realized that the DEM used for Titan2D model is noisy as it was merged

with TanDEM-X from Kubanek et al., (2013). During DEM reconstruction, TanDEM-X may produce random noise and grazing signal in complex topography area. In order to reduce the noise, we filtered and re-interpolated the DEM and re-run the model by using filtered DEM. Historically, volume with VEI 1, may produce debris avalanche/pyroclastic flows less than 5 km from the summit of Merapi (Voight et al., 2000). In our results, the maximum run out distance is about 4 km from the summit. We think that our results represents typical geophysical mass flow that occurs in Merapi.

Reviewer: My conclusion is that, even if the data are interesting, they have been already partially published. The calculation of the stability is not new (except that it is applied to Merapi), not detailed enough and, maybe, partially wrong (C/Cs and Fu). The study is focused on a very local problem: the collapse of a small part of a frozen lava dome by rain. The simulation of pyroclastic flows is based on a questionable assumption (a cold lava dome can create pyroclastic flow) and, the limitations of the model used and the results are not detailed enough. In the current state, I think that the paper cannot be published and it must be deeply reworked before publication.

Response : We appreciated the comments and suggestions from the reviewer and thank you very much for the work to improve the manuscript. We have deeply re-worked and re-analyzed the results and revised the manuscript based on the suggestions and comments from the reviewers.

| emissivity | distance | $T_{refl}$ | $T_{atm}$ | RH | Comp trans | Ext. trans | Ext. temp | Refer temp | Min (℃) | Max (℃) | Avg (℃) | RMSE (℃) |
|---|---|---|---|---|---|---|---|---|---|---|---|---|
| 0.95 | 300 | 0 | 20 | 45 | 0.91 | 1 | 0 | 20 | 7.3 | 137.7 | 34.7 | 0 |
| 0.96 | 300 | 0 | 20 | 45 | 0.91 | 1 | 0 | 20 | 7.3 | 136.7 | 34.4 | 1.04 |
| 0.95 | 1200 | 0 | 20 | 45 | 0.91 | 1 | 0 | 20 | 7.3 | 137.7 | 34.7 | 0 |
| 0.95 | 300 | 10 | 20 | 45 | 0.91 | 1 | 0 | 20 | 6.8 | 137.5 | 34.3 | 0.67 |
| 0.95 | 300 | 0 | 19 | 45 | 0.91 | 1 | 0 | 20 | 7.5 | 137.7 | 34.8 | 0.22 |
| 0.95 | 300 | 0 | 20 | 55 | 0.91 | 1 | 0 | 20 | 7.3 | 137.7 | 34.7 | 0 |
| 0.95 | 300 | 0 | 20 | 45 | 0.9 | 1 | 0 | 20 | 7.2 | 138.6 | 34.8 | 0.91 |
| 0.95 | 300 | 0 | 20 | 45 | 0.91 | 0.99 | 0 | 20 | 7.4 | 138.6 | 35 | 0.95 |
| 0.95 | 300 | 0 | 20 | 45 | 0.91 | 1 | 10 | 20 | 7.3 | 137.7 | 34.7 | 0 |
| 0.95 | 300 | 0 | 20 | 45 | 0.91 | 1 | 0 | 10 | 7.3 | 137.7 | 34.7 | 0 |

**Fig. 1.**

---

## Author Response (AR1)

Reviewer: This paper investigates dome instability at Merapi after the 2010 eruption. Although I agree that this kind of study is of primary importance to better assess the related hazards associated with the future occurrence of dome-collapse events at Merapi, I would only recommend this manuscript for publication in NHESS journal after major revisions. Firstly, the language used by the authors in the text is sometimes limited and confusing. Many paragraphs are not readable and/or comprehensible, too long and repetitive, and some even lack of meaning. I have outlined some of the main issues in the attached PDF but could not pay attention to every typo, grammar, repetitions, waffles and sentence structure problems. I would invite the authors to entirely revise some parts of the manuscript by taking into account the comments included in the attached PDF.

**Response: We appreciate this comment and made appropriate changes. We have revised and deeply re-worked the editing and language. We checked and corrected the grammar mistakes, deleted some typos and repetitive sentences, and re-phrased or deleted confusing paragraphs in order to improve the manuscript. A native speaker was proofreading the manuscript and found further language deficiencies that could be corrected.**

Reviewer: Secondly, the scientific part of the manuscript is somehow incomplete in some aspects. Although some of the issues related to dome stability are correctly described and discussed, some of the concepts presented in this paper lack of new innovative ideas.

**Response: Accepted comment and changes made. We further clarified the novelties of the work. We conducted the first geomorphology, thermal and structural mapping of the southern dome at Merapi (mentioned in Introduction page 3 L7-8). We are able to identify sub meter fractures and quantify the structural pattern of the unbuttressed dome sector in detail. The fractures are actively degassing as identified by our thermal camera. The geomorphology, structure, and thermal datasets were then used to investigate a potential hazard using two methods, factor of safety for the first time (see page 2 L31-32) and Titan2D. Application of the factor of safety calculation from Simmons et al (2004) and Titan2D (previously used at Merapi by Charbonnier and Gertisser (2009; 2012) we are able to evaluate the hazard arising from this unstable dome sector. We therefore think that the paper contains innovations justifying publication, which we could now further clarify in the revised version.**

Reviewer: The authors should rather focused on the recent structural features that developed in the entire summit area, including the crater rim and upper part of the cone, and not only the post-2010 lava dome.

**Response: Accepted comment and changes made. We re-analyzed the geomorphology, structures, thermal distribution and alteration area at the summit of Merapi (see section 3.1, page 6 L30). We also re-calculated the factor of safety for the south and the west flanks (see section 4.3 page 11 L12-27), which are progressively altered and thus experience structural instability in the near future.**

I think the recent 2018 explosive events should be taken into account, especially for the results presented in figure 5 and 6 showing the link between water percolation and slope failure as well as the deep structure of the summit area; but also for the discussion about flow hazard assessment, given the high potential of larger hazard associated with a larger scale event!

**Response: comment accepted and changes made. We added a short discussion and relationship of the 2018 explosions in section 4.2 (see page 11 L4-10). As the dome is also subjected to hydrothermal alteration, we assess and re-calculate the flanks instability by using factor of safety of Swedish slice/Fellenius method, which was also suggested by second reviewer (see section 4.3). We further explained parameters and forces that influence the factor of safety calculation more detail in the revised version. Discussion and limitation of each parameter is also added in the revised version.**

The authors also completely misunderstood the use of varying basal friction angles associated with different flow volumes, as explained in details in Charbonnier and Gertisser (2012). I suggest them to read carefully the paper and change the basal friction angles accordingly. An explanation about why Titan2D cannot model surges is also lacking...

**Response: Thank you very much for the suggestion. We agree that basal friction angles should be considered with care, and we understand the limitations. We have read carefully the paper and change the basal friction angle according to Charbonnier et al (2012) (see table 2 and page 6 L18-23). We added a short limitation of Titan2D related to pyroclastic surge in the discussion (see section 4.1 page 10 L5-10).**

Finally, the discussion section is badly written and should focus more about the results shown in section 3, particularly the structural and geomorphological data obtained, rather than just conversing on dome collapse hazards at Merapi. This could considerably straighten some of the interesting results obtained in this study by justifying the use of some new innovative techniques (TLS, SfM) to solve the issues outlined in the previous sections.

**Response:** Thank you for the critical comments. We thoroughly rewrote the discussion section. We added more detail about the geomorphology and structure at the summit (see section 3.1 and section 4.2). After the climactic eruption in 2010, the morphology of the Merapi summit has changed dramatically. As we describe in the revised version, previous study concerned the dramatic topographic changes (Kubanek et al., 2013) by comparing satellite radar data before and after the eruption and calculating the volume of the 2010 eruption. Our datasets now provide a much better resolution of the changes occurring within this newly developed crater, and could describe more realistic condition of the current morphology and structure of the Merapi summit. We now outline this aspect in our revised manuscript.

In addition, geomorphology and structural mapping at a steep-sided dome building volcano such as Merapi is challenging. For this we have realized a method linking lidar and drone based mapping. Combination of TLS and SfM is very promising and provides high quality datasets to identify sub-meter fractures and slope changes. This is the first study to map geomorphology and structure at the Merapi summit by using these techniques. Advantageous and limitations toward these techniques are further discussed in the revised version (see section 4.1 page 9 L1-8). To further improve the flow and organization of the discussion section, we let it proofread by native speaking colleagues.
Review of the manuscript: Dome instability at Merapi volcano identified by drone photogrammetry and numerical modeling by Herlan Darmawan, Thomas R. Walter, Valentin R. Troll, Agus Budi-Santoso

Reviewer:
Dear Editor,
The paper of Darmawan et al. focuses on the potential destabilization of the frozen dome of Merapi volcano by rain. It first calculates the morphology of the summit dome of Merapi volcano, then the dome stability and, finally, the pyroclastic flows that can result from a collapse. My review follows the structure of the manuscript: title, morphology calculation, stability and pyroclastic flow modelling.
Title: The title must be changed to indicate that the studies focuses on the effect of rain andis not a comprehensive study of all the mechanisms than can lead to the collapse of active domes, as suggested by the current title.

**Response: We appreciate this comment and made appropriate changes. We have changed the title to: "Structural weakening of the Merapi dome identified by drone photogrammetry after the 2010 eruption."**

Reviewer:
Morphology: The data obtained from Terrestrial Laser Scanning (TLS) and from drone are impressive and shows the power of such methods for calculating the morphology of dangerous areas. For the topography, most of the data were already published and the novelty here is to extract the topography profile as well as the fumarole locations from visible images.

**Response: Yes, part of the data is published earlier. In this new study we added new close up views of a photomosaic generated from new drone overflights (see Fig. 2 and section 3.1). This new data adds further information about the location of fumaroles and hydrothermal alteration. Another very important new dataset is the use of high resolution thermal infrared maps (see Fig. 5). These were generated by a superzoom lens and image mosaicking. The results allow identifying precise positions of gas escape. This gas escape follows a structural pattern already inferred from optical data, and weakly expressed in the Terrestrial Laser Scanning results. Therefore, this paper presents a number of novel and innovative methods. In the revised version we improved this description and made the novelties further clear in the introduction (page 3 L7-8) and discussion (page 9 L5-8).**

Reviewer: Another novelty is to couple thermal images to confirm the fumaroles locations. My only doubt on this part is the temperatures accuracy given by the authors. According to variation of the atmosphere humidity, the composition of magmatic gas and the variable distance from the camera (it seems that a mean distance of 300 m is taken into account for the correction instead of the real distance, calculated with the DEM) and the pixel size, an error of the temperature of only 3°C seems very accurate. Could the authors give more details on how they have obtained this accuracy estimation? If not, it can be stated that the temperature is approximate, which is enough for the needs of the manuscript. Line 29 of section 3.1 must also be modified: as fumaroles activity is also related to rain, a punctual observation of more visible fumaroles can be related to seasonal changes and not necessarily to an increase of the activity.

**Response: Accepted comment and changes made. We follow the reviewers suggestion and describe the temperature as approximate/apparent (see results in section 3.2), which is indeed enough for the needs of the manuscript. Nevertheless we assess the temperature uncertainty following (Spampinato et al., 2011). By varying parameters of emissivity, distance, reflection temperature ($T_{refl}$), atmospheric temperature ($T_{atm}$), relative humidity (RH), computed transmission, external optics temperature, and external optics transmission, we could assess the uncertainty. The uncertainty was obtained by choosing one pixel in the same area, varying one parameter, and then calculating the RMSE (see table 1 in attached file). Based on the calculation, we found that increasing emissivity by 0.01 may influence the apparent temperature of 1.04°C. Other studies of the dome rock emissivity at volcanoes (Merapi, Carr et al., 2016) and Colima (Walter et al., 2013) suggested that the emissivity may be in the range of 0.95 and 0.98, therefore, we estimate that the uncertainty of the thermal pixel value is ~3°C. However, in order to improve the manuscript, we accepted the suggestion from the reviewer by clarifying that the temperature is approximate/apparent (see section 3.2 in the revised version).**

Reviewer:
Factor of safety: This section is essentially based on the work and the model of Simmons et al. (2004). The novelty is the application to Merapi. My main criticism is that it is not easy to understand the calculations that have been done, and that some formula are perhaps wrong.

**Response: Accepted comment and changes made. Following language proofreading, some of the unclear phrasing might already improve the clarity of the text. Furthermore, we improve the description of the safety equation (FS). Factor of safety is widely used to calculate slope instability and it is calculated by dividing resisting forces to driving forces that acting on a failure plane (α) (see in Introduction page 2 L25-32). The conventional model such as Slice, Swedish/Fellenius's method are commonly used to calculate slope instability.**
**However, in an active lava dome, some additional forces may influence the resisting and driving forces. The FS equation used by us is based on Simmons's work, which aims to calculate dome instability during intense rainfall. Rain water may build up gas (Fu), vaporize the water (Fv), and add water forces (Fw). The FS from Simmon et al. (2004) considers the uplift force that may reduce the resistance force (W.cos(α)) and the water and vaporized water forces that may add the driving force (W.sin(α)). Therefore, we do not think that the equation is wrong, but we improved the text flow.**
**As the current Merapi lava dome is influenced by degassing and rainfall activities, we used the FS equation from Simmons et al. (2004) to estimate the failure plane inclination**

(α), therefore, we were able to quantify the volume of source collapse. In order to make the equation more understandable, we clarified each parameter in the method (section 2.2) in the revised manuscript. We also re-calculate the factor of safety by using Swedish/Fellenius method to compare our FS results in the revised version (see section 4.3).

Reviewer: It must be explained why the authors focus on a small portion of a frozen dome. All the summit, including the frozen dome, is cut by fractures. The crater flanks are very steep and can also collapse. The whole summit must be studied for a complete study of destabilizations.

**Response**: **very good comments. Obviously our description of the horseshoe shaped fracture and the instability tests applied for a particular dome sector were not clear. We focus on a small portion of the Merapi dome because we find a structural weakening due to hydrothermal alteration at the southern part of the lava dome. This structural weakening is evidenced based on digital elevation models showing a horseshoe shaped crater (see section 3.1 and 4.2), a fumarole expression following this horseshoe shaped pattern, and degassing of hot fluids along such a horseshoe shaped fluid pathway. It has been explained in the introduction that hydrothermal alteration may weaken dome rocks (page 2 L12-13) and promote dome collapse. We improved the description of the horseshoe shaped fracture in the revised version (see section 3.1. page 7 L10-18).**
**However, the idea to assess dome instability of the whole summit is good and accepted. We added instability analysis at the western flank in the revised manuscript as this area is also subjected by progressive hydrothermal alteration (see discussion in section 4.3). Previous studies also suggest that the dome collapses were dominantly to the west-south west direction in 1900's (Voight et al., 2000).**

Reviewer: Even if the safety model has been developed by others, the reader needs some information to understand what has been done (even briefly). Among the questions: what are the basis of the model?

**Response**: **We introduced the factor of safety model in the revised version to describe the application of this model to the readers (see page 2 L25-32). As mentioned before that the factor of safety is generally used to assess slope stability. The model is calculated by comparing resisting force to driving force that acts on a failure plane. We added the basic concept of factor of safety in the revised version, compare our FS results from Simmons et al. (2004) to the FS results from Swedish/Fellenius's model (see section 4.3).**

Reviewer: Why is there a link between the depth of water percolation and the distance between fractures to the square? (this is probably related to the surface that supplies the fracture, but why to the square?).

**Response**: **The instability of the square is influenced by water percolation (d) and the fracture spacing (s). Sensitivity tests of these two parameters suggested that increasing fracture spacing slightly decreases the factor of safety, while increasing water percolation (d) three times may reduce the stability 0.16 to 0.27 (Simmons et al., 2005).**

Reviewer: Why the fracture widths are not taken into account in the percolation depth calculation?

**Response: a critical question, which also leads us to further improve the discussion section of the paper. The water percolation is calculated based on equation 1 that consider a fracture spacing (*s*) on the block and dome properties and neglect the fracture width parameter. The equation considers that the dome properties (temperature, heat capacity, and thermal diffusivity) have stronger control toward water infiltration than the fracture width (Fig. 6) (see section 4.1). Therefore, fracture width is not necessarily to be taken into account in the percolation depth calculation.**

Reviewer: The "forces" must be more clearly explained and I recommend to expand and to detail the scheme of Fig. 6. The formulation of Fw and Fv are correct but it needs explanations: why a coefficient 0.5?Explain why, to calculate the force of the volcanic gas Fv, the density of the liquid and not of the gas is used (I have understood only by reading related papers).

**Response :We added detail explanation of forces that influence the FS calculations in the revised version (see page 5 L25-30). Fw and Fv are water and vaporized water forces, respectively. The coefficient of 0.5 is to calculate the geometry volume of Fw that influence the block (see Fig. 6). In general, force can be expressed by multiplying mass and acceleration/gravity acceleration (F = m. g, where m =v×ρ, so F = v×ρ×g). In the equation, it is mentioned that:**

**$Fw=(0.5×d^2×cos(α))×ρ_w×g.$**

**We inferred that the 0.5 is used to calculate the geometry volume of Fw that influence the block stability (see Fig. 6 where Fw are represented as triangle prism object).**
**However, we considered to re-calculate the volume of Fw in the revised version by using DEM and cross section profile to minimize error calculation and to give realistic parameter. We used the density of water to calculate Fv as Fv represents the force of vaporized water from rainfall that interacts with hot dome interior.**

Reviewer: The "forces" W and F are not real forces (in N) but forces per meter in (N/m). It might be called a force but after being defined correctly.

**Response : accepted comments. We corrected, converted all parameters in SI units and re-calculated the Factor of Safety in the revised version (See Table 1 and 2).**

Reviewer: In Equ. 2, C must be a force per meter. Is it the same as Cs, in tab. 1, both called "cohesive strength" but with a unit of stress (Pa)? The authors used the formulation of Simmons et al. and reproduce a probable typo in the formula (Eq. 1 of Simmons et al., 2004): Cs was probably C*s (in N/m in this case). Are the results obtained with a correct formula or with C instead of C*s? Because s = 100 m, using C instead C*s will significantly change the results.

**Response : Accepted comments.We corrected Cs to C\*s (see eq. 2), converted all parameters in SI units, re-calculated the FS from Simmons et al.(2004) and compared the FS results from Simmons et al to the FS calculation based on Swedish/Fellenius's methods.**

Reviewer: I cannot understand what is Fu and how it is calculated. If it is the water pressure at the base of the dome, the "force" must equal the pressure at the base of the fracture multiply by the dome surface, and it must be: Fu = d*cos(a)*rhow*g*s (neglecting the gas density). Where does the coefficient 0.5 come from? A progressive pressure decrease to the front? Why is it called the "uplift force from the volcanic gas", if it is related to the pressure of the liquid water only?

**Response : The uplift force (Fu) is produced by water vapor and volcanic gas that released upward through the fracture to atmosphere (see page 5 L28-30). As mentioned before that the coefficient 0.5 is used to calculate the volume of Fu that influence the block stability (see Fig. 6 where Fu are illustrated by triangle prism object).**
$$Fu=(0.5\times d^2\times cos(\alpha)\times s)\times\rho_w\times g.$$
**However, we considered and re-calculated the volume by using our DEM and we used the density of gas as suggested by reviewers (see page 5 L29-30).**

Reviewer: Once all these points will be fixed / clarified, the other point is the sensitivity of the model. The authors say that the "calculation requires careful parameter justification" (section 5). As several parameters seem estimated roughly, other graphs like that of Fig. 6 are needed to explore the model sensitivity to the cohesive strength, the temperature, the volume rate of the rain, the fracture spacing, etc on the stability. I think that friction angles of 60_ are not realistic and it can be replaced by a friction angle of 20° and 40°.

**Response : accepted comments and suggestions. In the revised version, we clearly described the limitation of the FS method and the sensitivity of the parameters (see section 4.1). We re-calculated the factor of safety by using friction angle of 25° and 45° according studies from Simmons et al (2005) and Husein et al (2014) (see page 6 L5).**

Modelling of pyroclastic flows:

Reviewer: The volume that can collapse in this manuscript is small and the lava dome is cold for several years (except because it transmits the temperature of the gases). In this case, why do the authors expect the genesis of pyroclastic flows? Even pyroclastic surges are evoked (4.1, p. 11, line 1-2). This assumption seems surprising and needs to be explained clearly. Works cited (e.g. Elsworth et al.) focus on an active and hot lava domes. The limitation of the numerical model used must also be presented. For example, the deposits of figure 7 are not compatible with pyroclastic flow deposits. They accumulate at the foot of the volcano forming piles more compatible with small rock collapse. Because the volume is small and the dome is cold, it is probable that a collapse will form a rock avalanche and not a pyroclastic flow but this must be explained and parametrized clearly.

**Response : Accepted comments. We decided to use the terminology of block and ash flow in the revised manuscript as even the outer carapace is cold, however, the dome interior is hot ~200°C based on our thermal camera data. In a case of southern block collapse, we infer that it may produce block-and-ash flow. However, we added limitation of Titan2D to model pyroclastic surges (see section 4.1 page 10 L3-10). We think that it is vital to assess the potential hazard zone in a case of the southern dome collapse as many cases suggested that hydrothermal alteration may weaken the rock and trigger a collapse. In addition, sand mining intensively occurs at the southern flank with radius of 5 km from the summit of Merapi. This is the reason why we assess the potential hazard of the southern Merapi dome sector. The idea to assess the potential hazard in this manuscript is vital for hazard assessment in the future.**

Reviewer: If the authors want to simulate pyroclastic flows, a long debate exists about the models and the approaches used for pyroclastic flows and, today, models of pyroclastic flows are not reliable enough to be presented without discussions and caution. In this context, two points seem very worrying to me: 1) the work recently published by Kelfoun et al (2017) on the same theme (numerical simulation of pyroclastic flows) and on the same volcano (Merapi, 2010) is neither cited nor discussed. It cannot be ignored even if the model seems to reproduce correctly a pyroclastic flow emplacement with a physics that differs from the physics of the present manuscript. 2) the references to Charbonnier et al (2013) are partial. Their work is cited to justify that Titan2D is a tool that makes good simulations of pyroclastic flows, avoiding discussions on the model limitations. However, even if they have shown positive features, Charbonnier et al. have also shown the limitations of the models. For example, they wrote: "Titan2D is not capable of reproducing the runout distances and areas covered by the actual events over the highly complex topography" (discussion, 5.3.2). Is it compatible with its use in the present manuscript? A model is never perfect and it is why the limitations of the approach and of the results must be clearly and honestly discussed. The remarks of SC1 on the interactive discussion are also significant: for example, is the simulation able to stop? If not, what criterion has been chosen?

**Response :Accepted comments. We still want to add block-and-as flow model because we find a structural weakening that may collapse in the future. The block-and-ash flow model is added to improve the hazard assessment and the flow of the manuscript, even though, the highlight of the paper is the finding of the geomorphology, structure, and hydrothermal alteration at the Merapi summit.**
**We used Titan2D to model the Pyroclastic block and ash flow as titan2D is well-validated to model granular avalanches over natural terrain (Patra et al. (2005); Pitman**

**et al. (2003). The suggestion to add the work from Kelfoun et al (2017) is accepted and added in the discussion (see page 10 L5-11). We agree that model is never perfect, therefore, we added more detail on limitation of the Titan2D to simulate debris avalanche in complex topography as also suggested by Charbonnier (second reviewer).**
**The remark of SC1 is also correct that Titan2D has limitation to stop the simulation. The simulation cannot perfectly stop, even it is reached the maximum time simulation. In order to fit realistic model, we extended the maximum time simulation up to 1 hour (see Table 2) which is long enough for rock avalanche duration and set validated/corrected friction angle parameter as this parameter control the run out and distribution of the rock avalanche (see Table 2 and page 6 L18-23).**

Reviewer: The quality of the DEM used, which seems to be very noisy, and the consequences on the results needs to be discussed too. Finally, given all the limitations of the approach and because the shape of the volcano has not changed from the last eruption, I wondered something similar to SC1: does the numerical model presented give results more confident than a rough estimation based on the past experience of Merapi's eruptions?

**Response :We realized that the DEM used for Titan2D model is noisy as it was merged with TanDEM-X from Kubanek et al., (2013). During DEM reconstruction, TanDEM-X may produce random noise and grazing signal in complex topography area. In the revised version, we merged our updated DEM with the DEM from Gerstenecker et al. (2005) as this DEM, so far, provides the best far field DEM of Merapi volcano and has no grazing effect in complex topography area, compared to the TanDEM-X.**
**Historically, volume with VEI 1, may produce debris avalanche/pyroclastic flows less than 5 km from the summit of Merapi (Voight et al., 2000). In our results, the maximum run out distance is about 4 km from the summit. We think that our results represent typical geophysical mass flow that occurs in Merapi.**

Reviewer: My conclusion is that, even if the data are interesting, they have been already partially published. The calculation of the stability is not new (except that it is applied to Merapi), not detailed enough and, maybe, partially wrong (C/Cs and Fu). The study is focused on a very local problem: the collapse of a small part of a frozen lava dome by rain. The simulation of pyroclastic flows is based on a questionable assumption (a cold lava dome can create pyroclastic flow) and, the limitations of the model used and the results are not detailed enough. In the current state, I think that the paper cannot be published and it must be deeply reworked before publication.

**Response : We appreciated the comments and suggestions from the reviewer and thank you very much for the work to improve the manuscript. We have deeply re-worked and re-analyzed the results and revised the manuscript based on the suggestions and comments from the reviewers.**
SC: The submitted manuscript provides a useful integrated study of drone-based geomorphological analysis and thermal infrared data collection to assess the stability of the dome of Merapi volcano. Water percolation within the dome is taken into consideration as trigger of dome collapses. The effort to provide a Factor of Safety is commendable. Although pyroclastic flow modelling is only a small portion of the research work illustrated here, to prevent this paper from being misleading, the authors should acknowledge the fact that there is still a lot of work to do before it is really possible to predict the mobility of pyroclastic flows.

**Response: Thank you very much for the comments. We agree to the points that the modelling is only a small portion of the work, and that there is still a lot of work to do before PDC can be predicted, and that especially the modelling technique used is limited. Therefore we are more careful with the interpretation of our result and inserted a critical discussion (see section 4.1. page 10).**

SC: I have a few important comments:

1) There is the need to mention the actual basal friction that the authors have chosen when running Titan2D: Coulomb, Voellmy or Pouliquen-Forterre, for example. If this is not done, it would be impossible to fully characterize the simulations.
   **Response: comment accepted. Titan2D uses the Coulomb friction to simulate geophysical mass flow over natural terrain (see section 2.3 L20). We have now added this information in the revised version.**

2) Please recognize in the text that Titan2D, as the name confirms, is a two-dimensional model whose results are adapted to a three-dimensional subsurface only later on by the software package.
   **Response: Accepted. We added this information in the introduction (page 2 L 33-34).**

3) It is also very important to disclose that, in Titan2D, the flows never stop and the computer operator has to introduce an arbitrary criterion to decide when the flows cease their motion and a deposit is formed [Ogburn and Calder, 2017]. The lack of acknowledgment of this shortcoming generates the false notion that the pyroclastic flow mechanics is understood.
   **Response: comment accepted, it is true that Titan2D will not technically stop in the end of simulation. In order to improve the result of block-and-ash flow, we maximize the computation time into 60 minutes (Table. 2) in the revised version. By using 60 minutes simulation time, we obtained a block-and-ash flow zone more realistic with run out distance of 3.6 (~4) km and inundation zone of 1.5 km$^2$. The run out distance and inundation zone in our result is very typical distribution of block-and-ash flow at Merapi for VEI 1 (volume $\leq$ 10$^6$ m$^3$).**

4) The main problem with Titan2D is that it ignores completely the granular nature of pyroclastic flows. This is in contrast to the fact that block-and-ash flows are well documented worldwide to be dense granular flows of angular rock fragments [Nairn and Self, 1978; Saucedo et al., 2002]. It is therefore important to inform the readers that an effort is undertaken to understand how rock fragments dissipate energy when interacting among themselves and the subsurface within travelling flows [e.g., Cagnoli and Piersanti, 2015 and 2017]. Since the grain size strongly affects the mobility, it is important to state clearly the grain size of the simulated flows.

**Response: We appreciate this comment. Titan2D software not completely ignores the granular nature of pyroclastic flows/debris avalanches. The flows are assumed to be incompressible continuum and the interaction between grains-grains and grains-basal surface is solved by Mohr-Coulomb law (see Patra et al., 2005). To discuss the limitations of the models and efforts of studying rock fragments in granular flows, we inserted the suggested references (Cagnoli and Piersanti, 2015 and 2017) (see section 4.1. Page 10 L12-18).**

5) My previous comments boil down to two questions. Considering that, block-and-ash flows are controlled by gravity and topography, do you really need Titan2D to know: A) that dome collapses discharge their rock debris down the deep and narrow valley which the horseshoe-shaped crater morphed into and B) that deposits form at the base of the volcanic cone where a dramatic change of the slope angle occurs?

**Response: We appreciate this comment. Yes, it is necessary to define source collapse mechanism in Titan2D. Titan2D is able to model several collapse scenarios such as a single collapse, multiple collapses, a gravitational collapse, or a fountain collapse that produce radial debris avalanches. In our model, we define that the mechanism of the source collapse is a single block collapse which triggered by hydrothermal alteration and neglect gas overpressure. Therefore, we chose a single gravitational collapse scenario (flux model) and set initial velocity of 0 m/s (no gas overpressure) and volume of ~300.000 m³ (volume of delineated block).**

**The deposit and the flow mechanism of Titan2D simulation are controlled by coulomb friction. In order to obtain realistic model where the flow is controlled by topography and different slope, we applied material map, which integrated with DEM. We defined variation of coulomb friction based on slope variation (see Table 2). In order to clarify the mechanism of source collapse and the debris flow, we added detail description of parameters that control the source collapse and the variation of coulomb friction angles in the revised version (see section 2.3 page 6 L15-25)**